# A CRISPR-drug perturbational map for identifying compounds to combine with commonly used chemotherapeutics

Hyeong-Min Lee [1,10], William C. Wright[1,10], Min Pan [1], Jonathan Low[2], Duane Currier [2], Jie Fang[3], Shivendra Singh[3], Stephanie Nance[4], Ian Delahunty[4], Yuna Kim[1], Richard H. Chapple [1], Yinwen Zhang[1], Xueying Liu [1], Jacob A. Steele [5,6], Jun Qi [7,8], Shondra M. Pruett-Miller [5,6], John Easton [1], Taosheng Chen [2], Jun Yang [3,9] ✉, Adam D. Durbin [4] ✉ & Paul Geeleher [1] ✉

Combination chemotherapy is crucial for successfully treating cancer. However, the enormous number of possible drug combinations means discovering safe and effective combinations remains a significant challenge. To improve this process, we conduct large-scale targeted CRISPR knockout screens in drug-treated cells, creating a genetic map of druggable genes that sensitize cells to commonly used chemotherapeutics. We prioritize neuroblastoma, the most common extracranial pediatric solid tumor, where ~50% of high-risk patients do not survive. Our screen examines all druggable gene knockouts in 18 cell lines (10 neuroblastoma, 8 others) treated with 8 widely used drugs, resulting in 94,320 unique combination-cell line perturbations, which is comparable to the largest existing drug combination screens. Using dense drug-drug rescreening, we find that the top CRISPR-nominated drug combinations are more synergistic than standard-of-care combinations, suggesting existing combinations could be improved. As proof of principle, we discover that inhibition of PRKDC, a component of the non-homologous end-joining pathway, sensitizes high-risk neuroblastoma cells to the standard-of-care drug doxorubicin in vitro and in vivo using patient-derived xenograft (PDX) models. Our findings provide a valuable resource and demonstrate the feasibility of using targeted CRISPR knockout to discover combinations with common chemotherapeutics, a methodology with application across all cancers.

Almost all curative cancer treatments result from combinations of multiple chemotherapeutic agents. However, existing drug combinations are often insufficient to provide a cure and cause severe side effects. The development of improved combinations faces several challenges. Firstly, the drug combinatorial search space is astronomical, with, for example, all possible 2 drug combinations of only 600 drugs yielding 179,700 combinations (given by $600^2/2 - 600/2$). All possible 3 drug combinations of 600 drugs yields approximately 100

million unique combinations, far beyond what could be screened using conventional approaches, without even considering variable compound dosage and timing. This suggests that improved high-throughput strategies are needed to capture this search space. Secondly, navigating drug approval for two investigational drugs simultaneously presents additional regulatory barriers and safety considerations over single-agent approval, which is already a difficult process[1,2]. This suggests that the development of combinations with

compounds already in clinical use should be prioritized as this will present the fewest hurdles to achieving rapid clinical impact. Combinations with standard-of-care drugs could also improve patient outcomes by mitigating toxicities if drug synergies mean that similar anti-tumor activity could be maintained at a lower exposure to broadly cytotoxic chemotherapeutics[3,4].

Pediatric neuroblastoma represents a particularly pressing clinical need. Neuroblastoma is the most common extracranial pediatric solid tumor[5], and despite intense study, survival in high-risk patients has remained close to 50%[6]. In recent years, numerous clinical trials have been conducted testing single-agent targeted chemotherapeutics. Most of these clinical trials (generally conducted in the difficult-to-treat relapse setting) have not been successful[6], typically due to limited tumor response. This has been true even when preclinical and mechanistic evidence has been very convincing; for example, for ALK inhibitors in ALK gain-of-function neuroblastoma[7], or IGF1R inhibitors in IGF1R overexpressing tumors[8,9]. These examples are among the few recurrently mutated druggable oncogenes in this disease, which occur in only a fraction of patients. Thus, the results of these trials suggest that combinations of drugs eliciting synergistic effects need to be considered, particularly in the recalcitrant and relapsed settings, if there is to be any chance of making clinical progress in this hyper-aggressive disease. This approach has demonstrated promise in other highly aggressive pediatric cancers, with for example the synergistic combination of PARP inhibitors with DNA-damaging agents in Ewing sarcoma achieving complete responses in some relapsed patients[10]. Despite this, large-scale drug combination screening studies in pediatric cancers have never been reported.

Here, we have exploited recent observations that CRISPR knockout of many druggable genes mimics pharmacological inhibition of the protein encoded by that gene[11]. Considering this, we design a CRISPR knockout library targeting 655 known druggable genes, including 55 pan-essential positive control genes and 400 non-targeting gRNAs as negative control. We screen this library to identify druggable gene knockouts that sensitize cell lines to commonly used cancer drugs, providing a large increase in throughput over conventional drug-drug combination screening approaches. Leveraging the resulting dataset, we propose therapeutics to combine with doxorubicin, topotecan, cisplatin, and the experimental proteolysis-targeted chimaera (PRO-TAC) agent JQAD1, which we show are effective using in vitro and (for doxorubicin) in vivo experiments. Overall, this resource provides a map for the discovery of chemotherapeutic combinations and demonstrates clinical potential in a disease of high clinical need.

## Results

### Design of CRISPR-drug perturbational screen and selection of cell lines

We first designed a targeted CRISPR gene knockout library (Supplementary Data Table 1) against 655 druggable genes, targeting each gene with 6 unique gRNAs (see Methods). Our list of druggable genes was based on Behan et al.[12] (Supplementary Data Table 1). Using Dep-Map data[13] as a reference, we removed genes with an expression of <0.1 $\log_2(TPM + 1)$ across the 10 neuroblastoma cell lines used in our screen.

We then set out to screen this gRNA library against a panel of 18 cell lines treated separately with each of 8 different drugs, or vehicle-treated control. The resulting relative abundance of gRNAs targeting each of these 655 genes in the drug-treated vs vehicle-treated cells provides a readout of whether target gene knockout sensitizes a cell line to a drug (Fig. 1a; Supplementary Data Fig. 1a; see Methods for details). This indicates that pharmacological inhibition of this gene product could also present a viable combination with the anchor drug[14,15]. In total, this experimental design yielded 18 cell lines × 8 drugs × 655 gene knockouts = 94,320 total unique combination-cell line pairs. The specific 8 drugs were doxorubicin, cisplatin, phosphoramide mustard (PM, the active metabolite of cyclophosphamide), etoposide,

topotecan, vincristine, and all-trans retinoic acid (standard-of-care neuroblastoma drugs that are all also used broadly to treat many cancers), and the PROTAC JQAD1[16], which is an EP300 degrader in preclinical development. We screened our gRNA library in 18 Cas9 stably expressing cell lines, 10 of which were neuroblastoma cell lines, and 8 of which were non-neuroblastoma cell lines (Fig. 1b, Supplementary Data Table 2). These included 4 cancer cell lines (from melanoma, Ewing sarcoma, rhabdomyosarcoma, and colon cancer) and 4 cell lines generated from normal tissues, specifically GM12878, a lymphoblastoid cell line, AC16, a cardiomyocyte cell line, BJ-TERT immortalized fibroblasts, and HEK293T cells[17]. The use of non-neuroblastoma cell lines in our experimental design provides valuable information in its own right, but also serves as a statistical out-group to evaluate the specificity of drug combinations. This provides a baseline to understand whether drug-CRISPR combinations are selectively lethal to neuroblastoma cells, or are simply broadly cytotoxic, which would likely reduce the chance of achieving a therapeutic window in patients. Neuroblastoma cell lines were chosen that already had prior exome-wide CRISPR-cas9 screening in addition to dense genomic and perturbational data available in the Cancer Cell Line Encyclopedia and DepMap. We used this information to nominate cell lines that cover the highest clinical need, including 5 cell lines with *TP53* mutations, which are enriched at relapse[18,19], and 4 mesenchymal-like cell lines, characterized by a gene expression program associated with chemotherapeutic resistance in neuroblastoma[20–22] (Fig. 1b, Supplementary Data Table 2).

### Knockout of known drug-target genes are the top hits for drug resistance

We were interested in assessing the validity of our 655 gene reduced representation gRNA library and screening approach. Thus, we first performed a targeted CRISPR screen using this gRNA library in CHP-134 neuroblastoma cells treated with the chemotherapeutic CX-5461 and DMSO-treated controls, an experiment we have previously performed using the Brunello genome-wide gRNA library[23]. Our previous genome-wide screen identified that CX-5461 is a topoisomerase inhibitor, with a high affinity for TOP2B. Encouragingly, in this re-screen using our reduced representation library, the top hit for CX-5461 resistance was the primary drug target *TOP2B* ($RRA = 2.48 \times 10^{-9}$; Note: robust ranking aggregation (*RRA*) scores are the default statistical measure from MAGeCK, the de facto computational tool for CRISPR data analysis, see Methods), followed by the secondary drug target *TOP2A* ($RRA = 8.46 \times 10^{-9}$; Fig. 1c, d; Supplementary Data Table 3). We also recovered the top sensitizing knockouts, *ATM* and *TOP1* (Fig. 1c, d; Supplementary Data Table 3). Thus, technical replicates screened using our reduced representation targeted CRISPR gRNA library delivered results consistent with a widely used genome-wide library, supporting the robustness of our approach.

Since five of the eight drugs that we screened using the targeted library have known direct protein targets, we next performed an analysis treating these targets as built-in positive controls to assess the validity of our results. Specifically, doxorubicin and etoposide directly target TOP2[24,25]; all-trans retinoic acid directly interacts with the retinoic acid and retinoid receptor family (RARA, RARB, RARG, RXRA, RXRB, and RXRG)[26]; topotecan targets TOP1[27] and JQAD1 degrades EP300 by selective recruitment of the E3 ligase receptor cereblon (CRBN)[16]. Cisplatin, PM, and vincristine act on DNA or microtubules and were not included in this analysis. Thus, for each of these 5 drugs, we calculated the mean resistance RRA scores (*Z* score normalized) across the screens performed in all 18 cell lines. Encouragingly, in all cases, loss of the known protein target of each of these drugs was identified among the top resistance mechanisms, and in the cases of retinoic acid, JQAD1 and topotecan were ranked #1 (Fig. 1e–i, see Supplementary Data Table 4 for all RRA scores, positive and negative, across the entire screen). These resistance mechanisms were

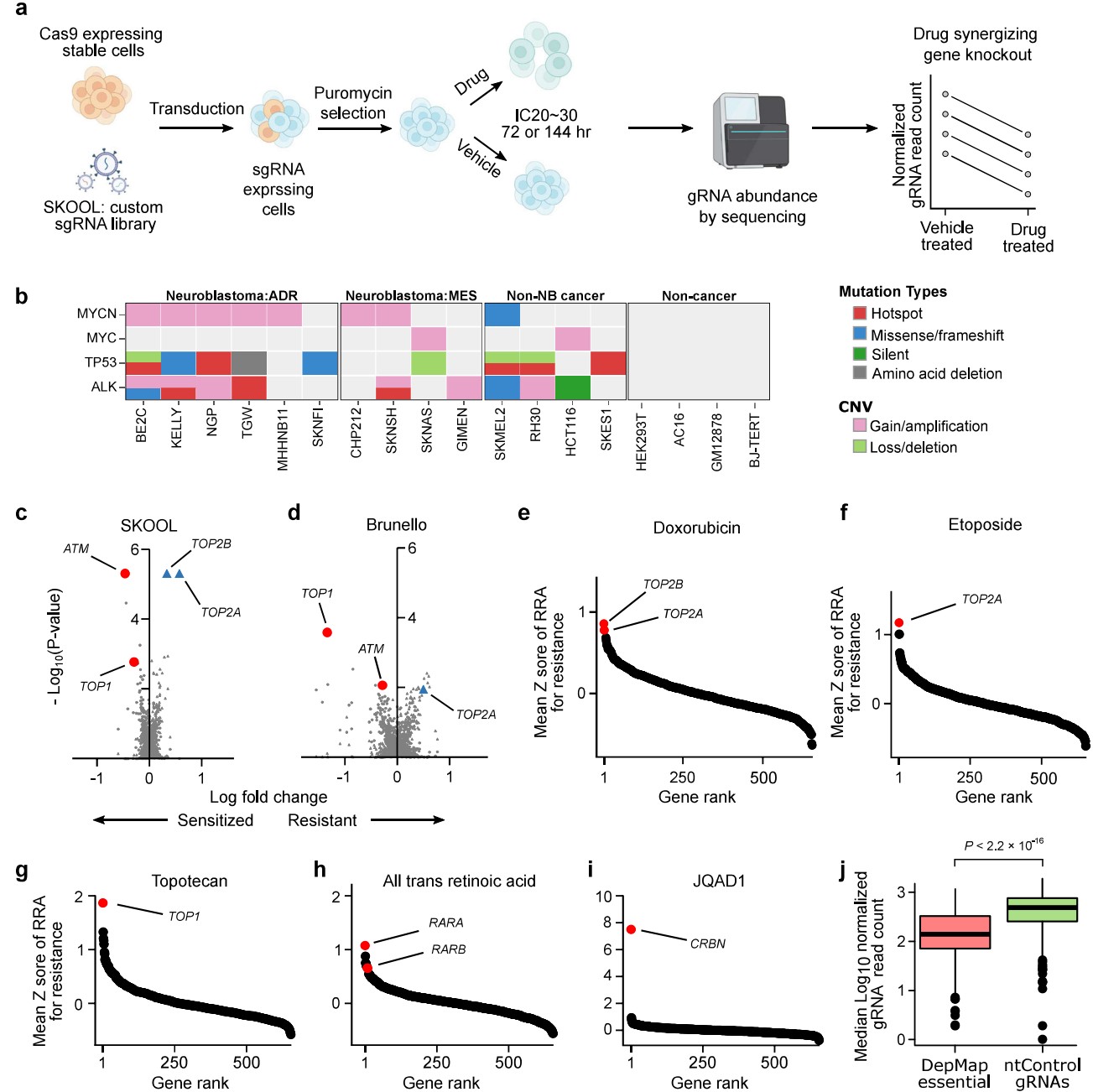

**Fig. 1 | A CRISPR knockout library, targeted to druggable genes, is a viable strategy to prioritize potential synergies with commonly used chemotherapeutics. a** Simplified schematic of CRISPR-sensitizer screens (detailed schematic available in Supplementary Data Fig. 1a). **b** Summary of genomics features of cell lines used in the screen. "ADR" refers to predominantly adrenergic neuroblastoma cell lines and "MES" to predominantly mesenchymal cell lines[20,22,92]. Mutation profiles were obtained from DepMap (details in Supplementary Data Table 2). **c** Volcano plot showing the log normalized gRNA fold change (LFC; $x$ axis) and $P$-values ($y$ axis) for each gene knockout in a CX-5461 vs DMSO control treated CHP-134 neuroblastoma cell line. Results were obtained using our 655 gene reduced representation library "SJ KnockOut nOn-Lethal (SKOOL)". Positive controls, the known sensitizing knockouts (*ATM* and *TOP1*) are highlighted as red circles and known resistance knockouts (*TOP2A* and *TOP2B*) are highlighted as blue triangles ($P$-values calculate using MAGeCK). **d** Like (**c**) but for results obtained for the same genes using the genome-wide Brunello library. **e**–**i** Waterfall plots showing the gene rank for resistance of the direct drug protein targets in our dataset. Genes are ranked by the mean $Z$ scores across all 18 cell lines of their RRA score for resistance. **j** Boxplot showing the median $\log_{10}$ normalized gRNA read count from all screens in all cell lines ($y$ axis) for $n = 400$ non-targeting (nt) control gRNAs (green box) and $n = 326$ gRNAs targeting 55 pan essential genes defined by DepMap ($P$-value from $t$-test). In all boxplots, the center line represents the median, the bound of box is upper and lower quartiles and whiskers are 1.5× the interquartile range. Source data are provided as a Source Data file.

recovered despite our screens being performed in $IC_{20}$-$IC_{30}$ drug concentrations (Supplementary Data Fig. 1b, c), a dose range suited to identifying drug-sensitizing knockouts, rather than resistance mechanisms (see Methods).

Finally, in every screen we included 326 gRNAs targeting 55 DepMap-defined pan-essential genes, as well as 400 non-targeting gRNAs as additional positive and negative controls respectively. As expected, the pan-essential gRNAs strongly decreased cell viability across the entire screen, but the non-targeting gRNAs did not (Fig. 1j, $P < 2.2 \times 10^{-16}$ from Wilcoxon rank sum test; see Supplementary Information for sequencing and QC metrics; Supplementary Data Table 1 for library sequencing details). Collectively, these

analyses provide strong evidence of the integrity of the screening data.

## Systematic trends in the CRISPR screening data

While these screens recapitulated expected resistance mechanisms and identified promising combinations (next subsections), the size of this dataset also allowed us to study genetic perturbational effects on chemotherapy response at a larger scale than previously possible. We used t-SNE[28] and UMAP[29] to cluster the data, revealing an interesting trend, with the data primarily clustered by cell line, rather than by drug or outgroup status (Fig. 2a; Supplementary Data Fig. 2a–d; UMAP parameters were selected using an automated Monte Carlo approach, see Methods). This suggests a proportion of, but not all of the signal in the data (see subsequent sections), may be cell line specific. We investigated this by a simple and interpretable orthogonal analysis, calculating the number of cell lines crossing a nominally significant RRA < 0.05 for each drug (Fig. 2b and Supplementary Data Fig. 2e) and for each cell line (Fig. 2c and Supplementary Data Fig. 2f). While there were no instances of a gene knockout sensitizing more than 7 of the 10 neuroblastoma cell lines to any given drug (Fig. 2b), there were several examples of CRISPR knockouts that sensitize a cell line to all drugs screened. For example, knockout of the anti-apoptotic gene *MCL1* sensitized SKMEL2 melanoma cells to all 8 drugs (Supplementary Data Table 4). These behaviors are consistent with previous observations that the effect of pairs of gene knockouts are often cell line specific[30,31], and that targeted drug combination effects are highly specific to cellular context[12,32]. Our results extend these observations to commonly used chemotherapeutics and suggest that caution should be exercised when extrapolating the results of drug combinatorial screens using small numbers of biological replicates (see Discussion).

## A hierarchical Bayesian modelcans identify gene knockouts that robustly sensitize multiple cell lines to standard-of-care chemotherapeutics

Because of our unique experimental design, and to maximize power to overcome cell line-specific effects, we developed a set of Bayesian hierarchical models to analyze these data—methods that can be easily implemented in future cohort-level screens. These models have been engineered to account for the uncertainty associated with drug-sensitization fold-changes, estimated from the differences in normalized gRNA read counts in drug-treated vs vehicle-treated control cell lines, which we treat as the model's outcome variable (see Methods).

   To identify the most potent drug-sensitizing gene knockouts across the entire dataset, we first applied this model to all 18 cell lines. Firstly, the resistance hits identified were similar to those identified from RRA scores (Supplementary Data Table 5) and were again consistent with the known drug targets, with the direct protein targets of etoposide, JQAD1, retinoic acid, and topotecan (*TOP2A*, *CRBN*, *RARA*, and *TOP1* respectively) all recovered as the #1 resistance knockouts when ranked by fold-change (Fig. 3a–h; Supplementary Data Table 5). Importantly, the model also revealed multiple drug-sensitizing gene knockouts. Many of the top sensitizing knockouts had a clear biological rationale and several have strong existing experimental evidence: These include *PARP1*, which was the #1 ranked hit (by fold change) for sensitizing cells to the TOP1 inhibitor topotecan (Fig. 3g; Supplementary Data Table 5). The synergistic interaction between TOP1 and PARP1 inhibitors has been validated in preclinical models[33] and represents one of the only standard-of-care drug synergies under active study in pediatric clinical trials[10]. This combination was originally investigated based on biological rationale because TOP1 causes single-stranded DNA breaks, which cannot be effectively repaired in the absence of *PARP1*, but in our data, this was identified without any prior biological knowledge. EP300 knockout was identified as the #6 ranked hit for sensitizing cells to the EP300 degrader JQAD1, suggesting that reduced EP300 levels potentiates this drug's activity

(Fig. 3d). PRKDC knockout was identified as the most potent sensitizer to doxorubicin (Fig. 3a), an association recently also reported in hepatoblastoma[34] and others[35,36], and a hit which we explore further in a subsequent subsection. *BCL2L1* knockout was the top-ranked sensitizer for cisplatin (Fig. 3b), #4 for phosphoramide mustard (Fig. 3e), #8 for vincristine (Fig. 3h), and #2 for topotecan (Fig. 3g). *BCL2L1* is an anti-apoptotic protein targetable by navitoclax, which has shown convincing synergy with several chemotherapeutic agents[37]. Combinations of navitoclax with cyclophosphamide[38], as well as regimens containing doxorubicin and vincristine[39], are in active clinical investigation[37]. The #2 cisplatin hit *DHFR* (Fig. 3b) is targeted by methotrexate, which is core therapy in osteosarcoma, combined in sequence with cisplatin[40], and #4 ranked cisplatin hit *CDC7* (Fig. 3b) is supported by existing in vitro results[41]. Doxorubicin has been shown to be synthetic lethal in combination with inhibitors of its #7 ranked hit *CDK1* (Fig. 3a)[42]. *MET* knockout was ranked #2 for sensitization to topoisomerase II inhibitors doxorubicin (Fig. 3a) and etoposide (Fig. 3c) and was ranked #7 for sensitization to cisplatin (Fig. 3b). MET overexpression has been widely implicated in chemotherapy resistance[43] and MET inhibition has already been reported to sensitize various cancer cells to doxorubicin[44,45] and cisplatin[46,47]. There is also evidence that the screens successfully identified other known resistance mechanisms beyond direct protein targets. For example, it was recently shown that mTOR inhibition represents a general chemoresistance mechanism[48], and this was identified as a top resistance hit for several of our screened chemotherapeutics (Supplementary Data Table 5). Overall, these results show that CRISPR-drug screens can recover known synergies, supporting their utility in identifying synergies with common chemotherapeutics in very high throughput (Supplementary Data Table 6).

## Gene set functional analysis reveals general chemoresistance mechanisms

We next wondered whether further systematic insights could be gleaned from our data by assessing functional relationships among the hits in the screen. To test this, we developed a gene set analysis approach tailored for these data, which compares a null distribution of fold-changes from randomly grouped non-targeting control gRNAs to the distribution of groups of functionally related genes using a Wilcoxon rank sum test (see Methods). We applied this approach to the drug vs vehicle-treated gene-level gRNA fold-change estimates for each drug individually and for the shared effect of the 6 DNA damaging agents considered jointly, performing these functional enrichment analyses at the level of (i) gene families, (ii) genes targeted by the same drug, and (iii) the molecular signatures database (MSigDB) "Hallmarks" gene sets, which represents a curated list of well-defined biological processes/pathways[49]. For a combined model assessing the 6 DNA damaging agents, groups of genes whose knockout is likely to slow cell proliferation were most clearly implicated in drug resistance (Fig. 3i; Supplementary Data Table 7). This included Hallmark gene sets such as G2M checkpoint ($FDR = 4.4 \times 10^{-3}$) and MYC targets V2 ($FDR = 2.9 \times 10^{-2}$; Fig. 3j; Supplementary Data Table 7). It is known that most common chemotherapeutics are more effective in fast-growing cells because DNA damage is much more likely to be induced during the cell cycle[50,51], and these results support this. Interestingly however, the EP300 degrader JQAD1 provides a compelling exception to this trend, with knockout of genes likely to support cell division surprisingly having the opposite effect and conferring drug sensitivity (Fig. 3i–k, $FDR = 2.5 \times 10^{-2}$ for "MYC Targets V1" for sensitization to JQAD1). Thus, knockout of some gene sets that confer chemoresistance, appear to confer sensitivity to JQAD1. This activity may result from JQAD1's ability to downregulate MYC family proteins[16]–however as an investigational compound, orthogonal drug activity compared to the existing standard-of-care is a desirable characteristic, as it is suggestive of potential to confer clinical benefit by independent action[52].

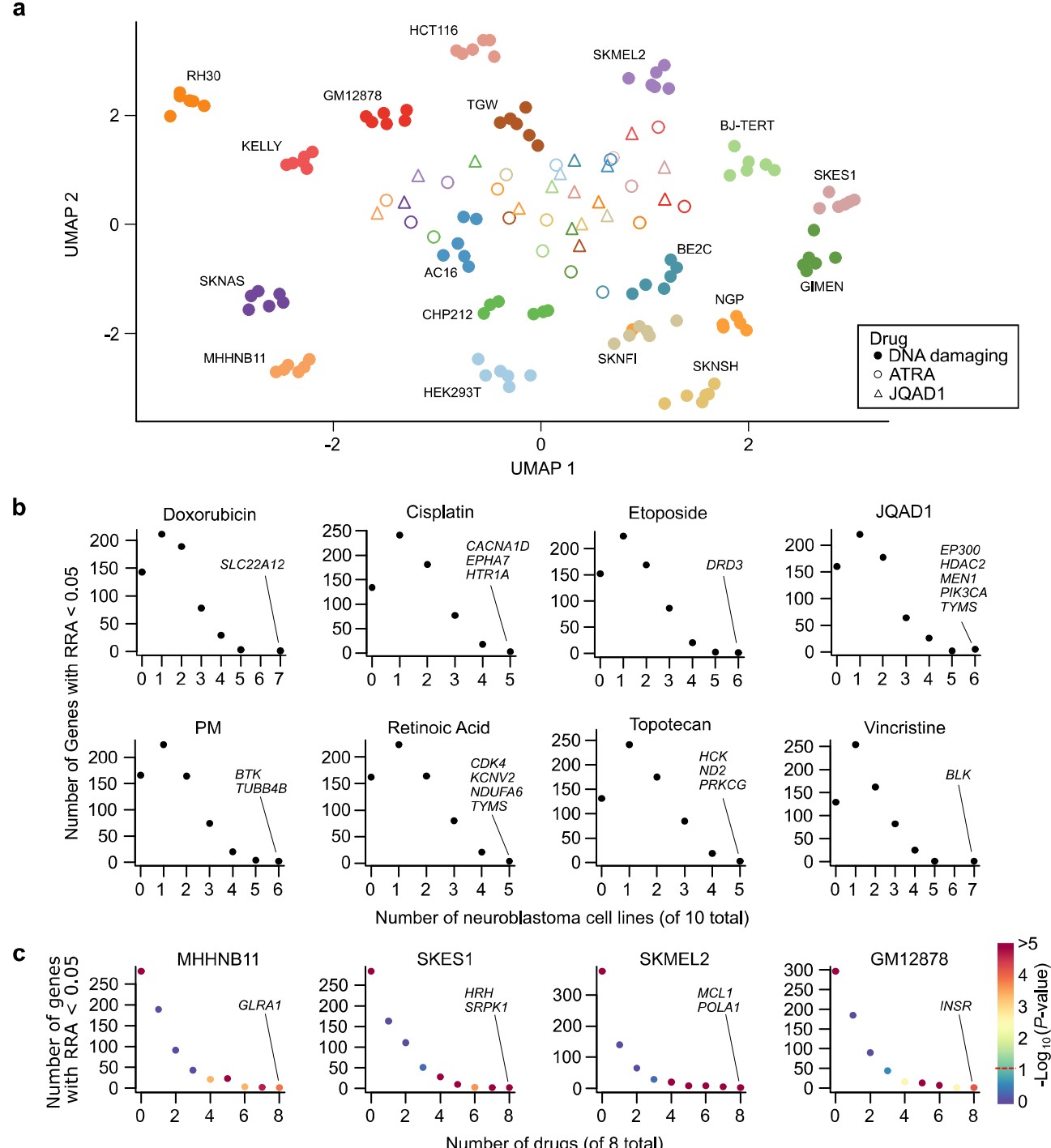

**Fig. 2 | Systematic examination of the screens identifies cell line-specific effects. a** UMAP representation of all 18 cell lines × 8 drugs = 144 total screens performed. Data are clustered based on fold-change of normalized read counts following each of the 655 gene knockouts profiled. Data points are colored by cell line and the symbol represents whether the cell line was treated with one of the 6 DNA damaging agents (solid circles; cisplatin, doxorubicin, etoposide, phosphoramide mustard, topotecan, or vincristine), all-trans retinoic acid (open circles), or JQAD1 (triangles). **b** Scatterplot showing the number of neuroblastoma cell lines (x axis) that are sensitized to some gene knockout (at RRA < 0.05; y axis) for each of the 8 drugs screened. Genes sensitizing the maximum number of cell lines to any given drug have been highlighted. **c** Scatterplot showing the number of drugs (x axis) that are sensitized to some gene knockout (RRA < 0.05; y axis) for each of the 4 different cell lines screened (all 18 cell lines are shown in Supplementary Data Fig. 2f). Genes sensitizing each of these 4 cell lines to all 8 drugs screened are highlighted. P-values were calculated by permutation of all RRA scores. Figure 2a–c. Source data are provided as a Source data file.

## Broadly chemo-sensitizing gene knockouts can be identified by using a Bayesian model that shares information across related drugs

In addition to the *MET* and *BCL2L1* examples discussed above, several hits could be identified in the analyses where genes appeared among the top sensitizers to multiple drugs that were independently screened. This is consistent with the mechanistic convergence of many chemotherapeutics on processes like DNA-damage, cell cycle, and apoptotic pathways (Supplementary Data Table 7). This is also consistent with the results of the functional enrichment analysis (Fig. 3i).

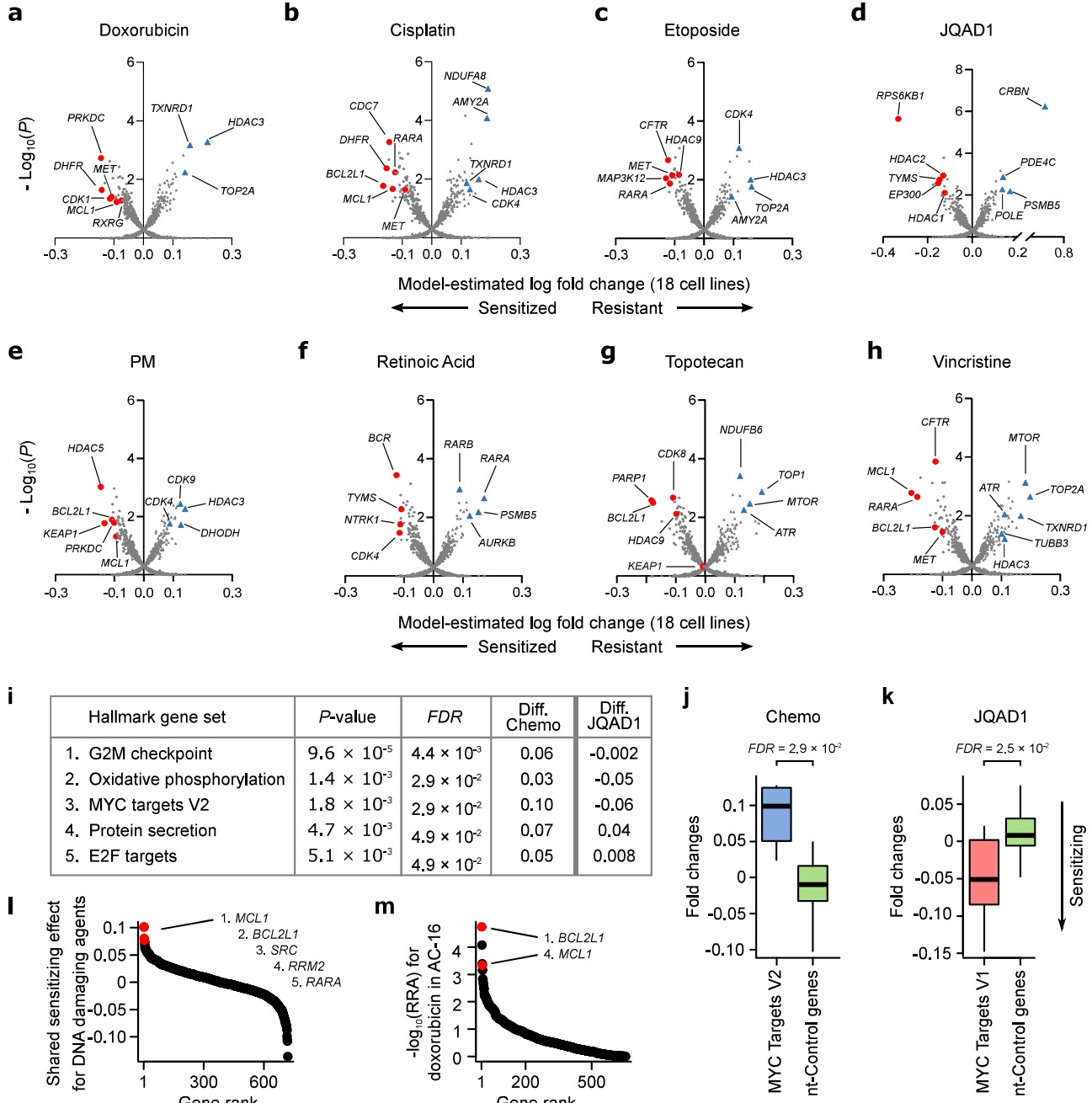

**Fig. 3 | Summary of drug sensitizing hits identified across the entire dataset.**
**a–h** Volcano plots for each drug showing the estimated mean drug vs mock-treated control normalized gRNA log fold change across all 18 cell lines screened (x axis) and the posterior probability this fold change value is different from 0 (y axis). These values were estimated by fitting our Bayesian Hierarchical model (see Methods) to the 18 fold-change values, and associated measurement uncertainty estimate, obtained for each gene knockout, in each drug. Lower fold change values imply a gene knockout has caused drug-sensitization, with top hits highlighted as red circles. Key resistance knockouts have been highlighted as blue triangles. **i** Top enriched mSigDB "Hallmark" gene sets from a joint model fit across the 6 DNA damaging agents screened. The values in the columns labeled "Diff." are the difference in fold change for the median gene in the gene set versus the median of the non-targeting control genes; negative values in these "Diff." columns imply knockouts of genes from this gene set are associated with drug sensitization. The final column shows the directionality of these hits for JQAD1, which are opposite of the 6 DNA damaging agents for the top 3 gene sets. **j** Boxplot showing mean

estimated drug vs DMSO normalized gRNA log fold changes (y axis) for genes annotated to the mSigDB's Hallmark MYC targets gene set (blue box, black bar = median (0.11)) vs non-targeting control genes (green box). For each gene, these mean fold change estimates were calculated across all 18 cell lines screened using a joint model that considered the 6 DNA damaging agents screened. (**k**) Like (**j**), but for JQAD1 (Hallmark MYC targets gene set (red box, black bar = median (−0.05)). **l** Waterfall plot ranking gene knockouts (x axis) by their shared sensitizing log fold change effect (y axis) across the 6 DNA damaging agents, estimated using our Bayesian hierarchical model. **m** Waterfall plot ranking gene knockouts (x axis) by -log$_{10}$ sensitizing RRA scores (y axis) in the AC-16 outgroup cell line. *BCL2L1* and *MCL1*, which are the top broad sensitizer genes in panel (**l**), are highlighted in red and sensitize this cardiomyocyte cell line to doxorubicin. In all boxplots, the center line represents the median, the bound of box is upper and lower quartiles and whiskers are 1.5× the interquartile range. Source data are provided as a Source Data file.

To formally identify these broad sensitizers in a statistically coherent framework, we extended our Bayesian model to share information across related drugs (see Methods), specifically the 6 DNA damaging agents which tended to co-cluster (Fig. 2a; doxorubicin, etoposide, cisplatin, topotecan, vincristine and phosphoramide mustard). These analyses revealed several broad sensitizers (Supplementary Data Table 8) with the top-ranked gene being *MCL1*, followed by the aforementioned BCL2L1 (Fig. 3l). This is interesting because *MCL1* has already been suggested as a potent sensitizer to several drugs in multiple diseases. These include the cytotoxic chemotherapeutics paclitaxel and docetaxel in breast cancer[53,54], reviewed in Bolomsky et al.[55].

These broad hits can also be used to highlight an interesting feature of our experimental design. *MCL1* knockout, while synergizing with multiple chemotherapeutics in neuroblastoma cell lines, also promotes cytotoxicity in our outgroup; for example, it also has strong synergy with doxorubicin in our cardiomyocyte cell line (Fig. 3m). Cardiotoxicity is an often-fatal side effect of doxorubicin treatment[17,56–59] and such a result suggests that this risk may be potentiated by MCL1 inhibition, an idea for which there is already some support in the literature[60]. Thus, while a joint model applied to these data can identify broadly synergizing knockouts, such targets may be a high risk for toxicity if selectivity is not considered. Motivated by this concept, we next introduce the idea of "selective drug synergy", where drug sensitization in our neuroblastoma cell lines is compared to our outgroup, assessing a gene knockout's potency *and* selectivity, which could help identify drug synergies more likely to have a therapeutic window in vivo.

### Gene knockouts selectively sensitize neuroblastoma cell lines to standard-of-care drugs

To test whether we could identify neuroblastoma-selective hits, we further extended our statistical models to handle case/control designs (see Methods), as well as probing covariates such as genomic features (Fig. 4a–I; Supplementary Data Table 9).

Because of the limited scale of existing screens, prior knowledge of the selective drug synergy landscape of neuroblastoma is currently almost non-existent. However, *RRM1* and *RRM2*, which we identified among the top hits sensitizing to etoposide and vincristine, may represent partial exceptions to this. Recently, combined RRM2 and CHK1 inhibition was shown to be synergistic in neuroblastoma xenografts owing to replicative stress due to stalled replication forks[61], a process in which TOP2, the target of etoposide, also plays a role[62]. However, most of the neuroblastoma selective synergies identified in our study are previously unreported. Interestingly, for some of the most potent drug synergizing knockouts identified across the full dataset, these results suggest their activity is stronger in the outgroup than the neuroblastoma group. *MCL1* and *BCL2L1* represent two examples of this, evident in the topotecan, vincristine, and cisplatin screens (Fig. 4e–g). This suggests that combinations of inhibitors of these targets with DNA-damaging agents should be approached with caution in neuroblastoma because while there may be potent synergistic activity in neuroblastoma cells, there is also strong potential for activity in normal cell types. In the context of other diseases, there is already some evidence that pharmacological inhibition of these targets in combination with DNA-damaging agents causes toxicities[63–65].

In general, while the number of screened cell lines and combinatorial perturbations in our screen is far larger than previous screens in neuroblastoma, it is still not large enough to confidently resolve the context specificity of many hits. However, there is some tentative evidence of orthogonal activity of some nominated combinations, with differential activity emerging on the background of, for example, the expression of mesenchymal-like genes—a putative

drug resistance state in neuroblastoma[20]. For example, the knockout of *PRKDC* appears to have a greater sensitizing effect in adrenergic neuroblastoma cell lines (Fig. 4j), whereas *KEAP1's* effect on topotecan (Fig. 4k), and *HDAC2's* effect on JQAD1 (Fig. 4l), are both sensitizing in mesenchymal-like cell lines (See Supplementary Data Table 9 (final tab) highlighting neuroblastoma relevant combinations for each drug). These observations provide tentative evidence that such context-specific synergies may exist for standard-of-care drugs in neuroblastoma, but broadly resolving these and deconvolving the various confounding factors will require detailed prospective experimental work. Thus, while we explore a selection of hits in detail below, we have also made the data and processed results available in a graphical web-based interface that can be used to motivate new studies dissecting the promising selective hits (available at https://stjude.shinyapps.io/CASAVA/).

### Synergies identified in high-throughput pooled CRISPR-drug screens translate to other genetic and pharmacological assays

We were next interested in assessing the robustness with which CRISPR-drug nominated synergies could be recapitulated with other in vitro and in vivo assays. We first assessed this using an orthogonal genetic perturbational assay, specifically shRNA knockdown of the CRISPR-nominated targets, for putative synergistic and non-synergistic interactions. First, we tested the knockdown efficiency of individual shRNAs against *PRKDC, HDAC2, KEAP1*, and *MET* to select the most efficient one (Supplementary Data Fig. 3a; see Methods). After selection, we knocked down four individual genes in a subset of cell lines (10 for *PRKDC*, 11 for *HDAC2, KEAP1*, and *MET*, a total of 43 knockdown experiments, Supplementary Data Fig. 3b) and treated with the corresponding drugs (IC$_{50}$ of doxorubicin in *PRKDC* knockdown, IC$_{50}$ or max 10 μM of JQAD1 in *HDAC2* knockdown, IC$_{50}$ of topotecan in *KEAP1* knockdown, and IC$_{50}$ or max 10 μM of cisplatin in *MET* knockdown). Encouragingly, we observed a strong positive correlation between the shRNA and CRISPR-based perturbations (Fig. 5a–d, Supplementary Data Fig. 3, Supplementary Data Table 10). Thus, the results from the pooled CRISPR screen could be replicated in a different low-throughput genetic perturbational assay.

Next, to test the translation of gene targets to pharmacological inhibition of protein products, we used dense drug-drug rescreening for these same hits (PRKDC/doxorubicin, KEAP1/topotecan, HDAC2/JQAD1, and MET/cisplatin). The drugs used to target the CRISPR-nominated genes were AZD7648 for PRKDC, dimethyl fumarate (DMF) for KEAP1, panobinostat for HDAC2, and cabozantinib (CAB) for MET. For each pair of drugs, we used high-throughput robotic handling screening in dense 10 × 10 matrices, with 1:3 dilution (10 doses from 10 μM to 1 nM for AZD7648, DMF, and CAB, and 10 doses from 0.01 μM to 0.5 pM for Panobinostat; Fig. 5e–j). In all cases, strong synergy was observed (Fig. 5k; Supplementary Data Figs. 4 and 5, Supplementary Data Table 11). Overall, the results suggest that CRISPR-drug screening results can translate to similar drug-drug combination screens, especially when the compound has a high selectivity for its target. We also performed similar dense 10 × 10 concentration drug-drug rescreens for each pair of standard-of-care neuroblastoma drugs (Cisplatin, PM, ATRA, Topotecan, Doxorubicin, and Vincristine; 15 total combinations) using the BE2C cell line and compared the synergy scores of the CRISPR-motivated compound pairs to synergy scores when standard-of-care drugs are paired. Remarkably, the CRISPR-motivated compound pairs were far more synergistic, as evidenced by much higher Zero Interaction Potency (ZIP) synergy scores ($P = 1 \times 10^{-4}$, Fig. 5l–m). Thus, it is likely that the enormous drug combinatorial search space contains drug pairings that can improve upon the existing standard-of-care combinations and high throughput drug-CRISPR screens represent a reasonable means to identify these.

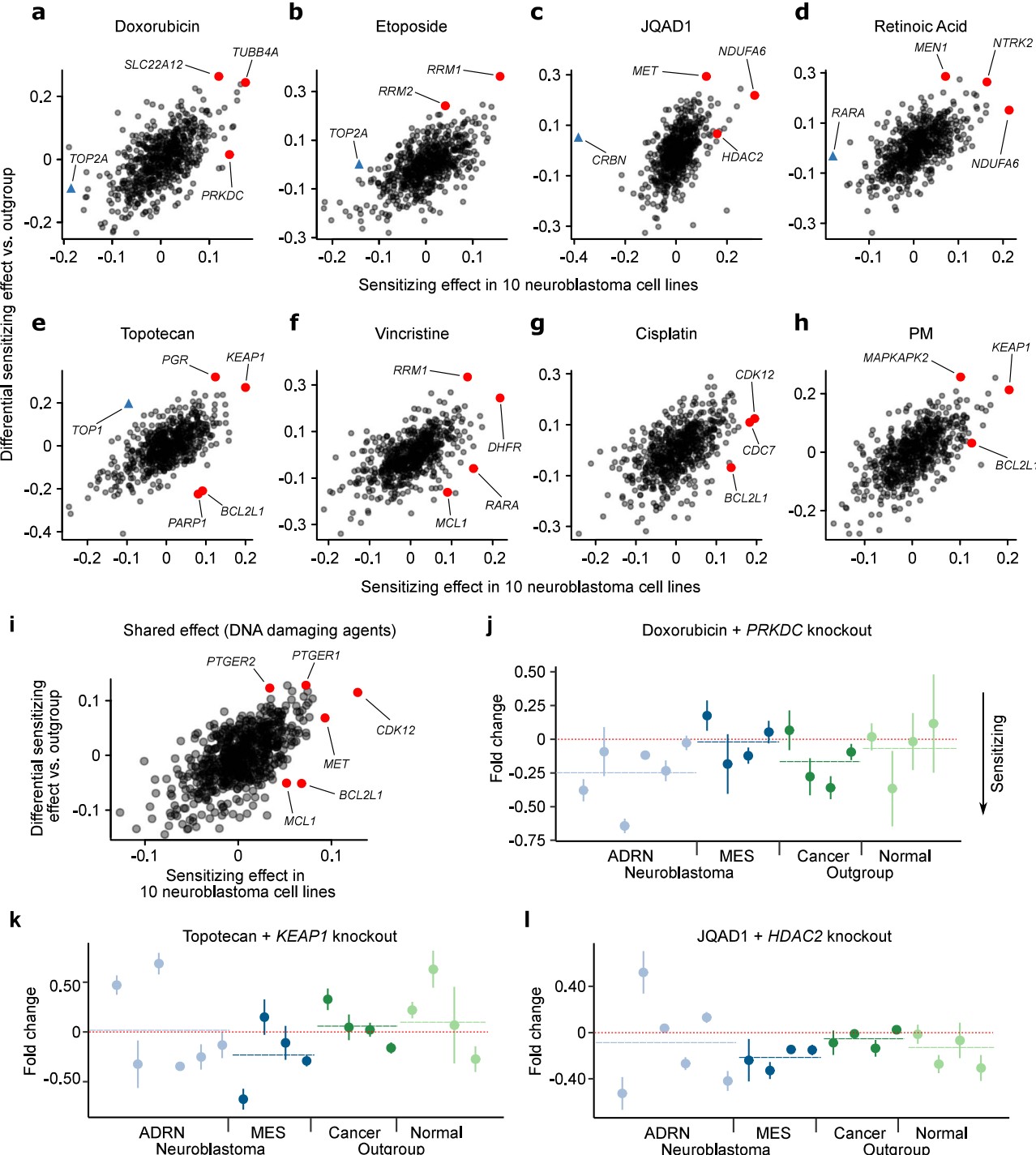

**Fig. 4 | Summary of differential drug sensitizing hits identified in neuroblastoma cell lines vs the outgroup cell lines. a–h** Scatterplots for each drug showing the estimated mean drug vs mock-treated control normalized gRNA log fold change across in the 10 neuroblastoma cell lines (*x* axis) and the differential sensitization effect between the 10 neuroblastoma and 8 outgroup cell lines (*y* axis). Higher values on the *x* axis imply greater sensitization in neuroblastoma and higher values on the *y* axis imply greater sensitization in neuroblastoma relative to the outgroup. **i** Like (**a–h**) but for the shared effect for the 6 DNA damaging agents,

estimated using our Bayesian hierarchical model. **j** Doxorubicin vs mock-treated control normalized gRNA log fold changes following *PRKDC* knockout (*y* axis) for all 18 cell lines screened. Lower values imply sensitization. Neuroblastoma cell lines are colored blue and outgroup cell lines are green. The order of the cell lines (*x* axis) is the same as Fig. 1b. Whiskers represent the standard error of the mean, estimated from the 6 gRNAs targeting each gene. **k** Like (**j**) but for topotecan and *KEAP1* knockout. **l** Like (**j**) but for JQAD1 and *HDAC2* knockout. Source data are provided as a Source Data file.

## PRKDC inhibition represents a mechanistically plausible combination with doxorubicin in high-risk neuroblastoma, with evidence of synergistic activity in vivo

The results above highlighted PRKDC inhibition as a particularly high-potential combination with doxorubicin in neuroblastoma. The

synergistic relationship between doxorubicin and PRKDC inhibition is plausible, as doxorubicin's primary mechanism of cytotoxicity is DNA double-strand breaks caused by trapping of TOP2 to DNA[66,67]. These breaks are repaired in part by the non-homologous end joining (NHEJ) pathway, where PRKDC plays a critical role. We performed several

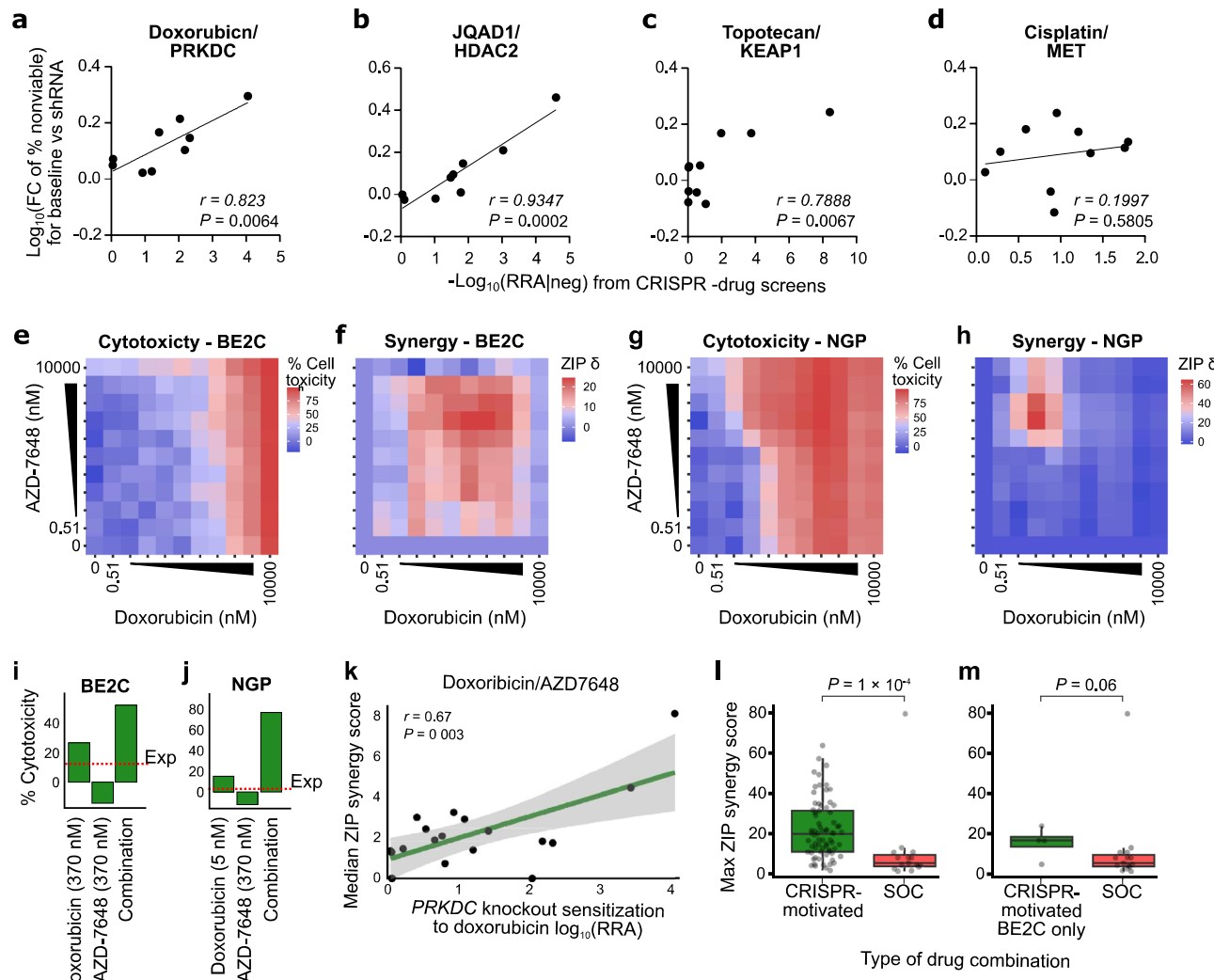

**Fig. 5 | Synergies identified in high-throughput pooled CRISPR-drug screens translate to other genetic and pharmacological assays. a** Scatterplot of the -log$_{10}$ sensitizing RRA scores from the CRISPR -drug screens (x-axis) against the -log$_{10}$ fold change of cell toxicity percentage upon knockdown of *PRKDC* using single shRNAs (y-axis) in 10 cell lines. For the shRNA knockdown, potency changes were estimated from the IC$_{50}$ of doxorubicin. See Supplementary Data Table 10 for source data (P-values were calculated by Pearson correlation). **b** Like (**a**) but *HDAC2* shRNA knockdown in 11 cell lines treated with JQAD1. **c** Like (**a**) but *KEAP1* shRNA knockdown in 11 cell lines treated with topotecan. **d** Like (**a**) but *MET* shRNA knockdown in 11 cell lines treated with cisplatin. **e** Heatmap matrices of percent cytotoxicity (1 - cell viability) in BE2C cells conferred by treatment with doxorubicin (x axis) and AZD-7648 (y axis). Each matrix represents the average of three independent experiments. **f** Heatmap matrices of synergy scores derived from cytotoxicity values in (**e**). All synergy scores δ were calculated based on the zero-interaction potency (ZIP) model. Combinations conferring synergy have ZIP scores of >0. **g** Like (**e**) but for the NGP cells. **h** Like (**f**) but for the NGP cells. **i** Bar plot showing the cytotoxicity values for the region of max synergy in the BE2C cells (panel (**f**)). The red dashed line shows expected cytotoxicity under additivity. **j** Like (**i**) but for the NGP cells. **k** Scatterplot of sensitization RRA scores in each cell line for doxorubicin sensitization by *PRKDC* knockout (x axis) versus overall synergy scores (y axis) from the AZD7648/doxorubicin drug-drug screen See Supplementary Data Table 11 for source data (P-values were calculated by Pearson correlation. The shaded band is a 95% confidence interval. **l** Boxplot of maximum synergy scores achieved in our complete set of CRISPR-motivated drug combination screens (green box, n = 72, median = 20.29) and standard-of-care (SOC, n = 15, median = 5.88) motivated drug combinations in the BE2C cell line (red box). P-values were calculated from a 2-sided Wilcoxon rank sum test. **m** Like (**l**) but showing only CRISPR-motivated drug combinations in the BE2C cell line (n = 4, median = 17.07). Figure 5a–m. Source data are provided as a Source Data file.

additional assays to assess the clinical tractability of this putative combination.

Upon DNA damage, PRKDC undergoes phosphorylation of residue ser-2056, causing a conformational change required for efficient end processing and DNA repair[68]. Thus, we performed western blots to examine the induction of phosphorylation at ser-2056 (pPRKDC) following doxorubicin treatment in the neuroblastoma BE2C and GIMEN cell lines (0.4 μM in BE2C, 0.03 μM in GIMEN – the approximate IC$_{50}$ of these cell lines used in all experiments in this section except where otherwise indicated; PRKDC knockout strongly synergized with doxorubicin in BE2C, but not GIMEN, in the CRISPR screen). The blots showed that single agent doxorubicin-induced pPRKDC in BE2C, but

not GIMEN, suggesting NHEJ was only strongly activated in BE2C, the comparatively doxorubicin-resistant cell line (Fig. 6a). Additionally, we confirmed the on-target activity of AZD7648, with single-agent treatment at 3 μM repressing pPRKDC in both cell lines (neither cell line was sensitive to single-agent AZD6748). Surprisingly however, in BE2C, the combination of doxorubicin and AZD7648 markedly increased pPRKDC over doxorubicin alone at 72 h (Fig. 6a), suggesting NHEJ activity had increased, an apparent contradiction worth further investigation. Thus, we next tested induction of apoptosis in each cell line using a luminescent caspase 3/7 assay and found that despite the high pPRKDC levels in the doxorubicin/AZD7648 treated BE2C, these cells also had the highest levels of apoptotic response (Fig. 6b,

$P < 1 \times 10^{-4}$ compared to doxorubicin alone). Unsurprisingly, the combination had little effect in potentiating apoptosis in GIMEN. We hypothesized these trends were likely due to much higher levels of DNA damage in the combination-treated BE2C cells. Indeed, at 72 h there was a 4-fold increase in γH2AX foci and a clear increase in overall DNA damage as estimated by a comet tail assay (Fig. 6c, d, $P < 1 \times 10^{-4}$ for doxorubicin treated vs combination treated in both assays, Supplementary Data Table 12). Interestingly, the number of γH2AX foci in BE2C was approximately 10 times higher than GIMEN when both cell lines were treated with an approximate $IC_{50}$ of doxorubicin (Fig. 6c; Supplementary Data Fig. 6). This is consistent with BE2C being a *TP53* mutant generally chemo-resistant cell line, requiring much higher levels of DNA damage to induce cell death programs. Since the relative use of DNA-repair pathways is also cell cycle-dependent, we further tested the combination effects when arresting cells in G0/G1 or G2 phases of the cell cycle, but neither could explain the increase in pPRKDC (Fig. 6e, Supplementary Data Fig. 7; see Methods). Finally, we performed a cell-based NHEJ activity assay, collecting data at 6 time points from 0 to 72 h. In BE2C, at later time points, NHEJ activity was higher in the combination-treated cells than in cells treated with doxorubicin alone and there was little evidence of induction of NHEJ in GIMEN (Fig. 6f, Supplementary Data Fig. 8). Rather, GIMEN induced the homologous recombination pathway (Fig. 6g). Thus, it seems likely that early DNA damage in the combination-treated BE2C cells leads (counterintuitively) to higher NHEJ activity in surviving cells at later time points, despite AZD7648 actively inhibiting NHEJ over the time course. It is plausible that some of these behaviors, including ser-2056 pPRKDC, emerge, at least in part, as a direct consequence of apoptosis[69]. Overall, these results suggest that differences in the reliance on the NHEJ pathway could predict the effectiveness of PRKDC inhibitors in combination with doxorubicin, and the main mechanism driving synergy is massive potentiation of DNA damage in cells dependent on NHEJ, which can be sufficient to induce cell death even in generally chemoresistant cells. Thus, potentiating doxorubicin activity in this context has the potential to address a clear clinical need[19].

As a final proof of principle, we tested this combination of doxorubicin and AZD7648 in vivo using mouse models of neuroblastoma. We used a pharmacologically relevant dosage of each drug based on existing pharmacokinetics data of doxorubicin[70,71] and AZD7648[36] (see Methods). In the first set of experiments, we implanted the BE2C cell line into NSG mice. We started drug treatment after tumor engraftment and growth beyond 100 mm³. We observed no significant effect of either AZD7648 or doxorubicin alone, but a clear reduction in tumor volumes when the combination was administered (Fig. 6h, Supplementary Data Table 13), suggesting synergistic activity against this cell line in vivo.

We conducted a second in vivo study, this time xenografting a human PDX model (NB14; previously established at St. Jude[72]) of high-risk *MYCN*-amplified neuroblastoma. AZD7648 had no detectable activity as a single agent, but, as in the previous experiment, strikingly potentiated the activity of doxorubicin, exhibiting strong control over tumor growth in all mice (Fig. 6i, Supplementary Data Table 13). Remarkably, this PDX is distinct from our cell line discovery cohort, suggesting broad relevance of this synergy in neuroblastoma and that this combination should be further evaluated in this disease. Overall, these mechanistic and in vivo experiments show that a large-scale CRISPR-drug perturbational map can be used to prioritize potential synergies with common chemotherapeutics and that these hits can be used to nominate drug combinations with clinical potential.

## Discussion

Drug combinations are required for essentially all curative cancer treatment strategies. However, the number of possible drug combinations is much larger than can be reasonably screened with existing approaches. The two largest existing drug combinatorial screens are the NCI's ALMANAC study[73], which screened the NCI60 cell line panel, and a very recent screen in 125 colorectal, breast, and pancreatic cancer cell lines[32]. Both studies screened a total of approximately 100,000 unique combination-cell line pairs (compared to 94,320 here). These previous studies employed moderately dense drug-drug combinatorial screening designs, where drugs were screened in 384 well plates in 3 × 3 or 2 × 7 grids of drug concentrations. Even with modern robotic handling, this approach is resource intensive, thus limiting the number of cell lines captured. Hence it is not surprising that most cancer types are absent from these studies, with, for example, almost no representation of pediatric cancer. To broadly capture the astronomical drug-drug combinatorial search space, resource-efficient approaches will need to be developed and deployed. Here, we performed large-scale targeted CRISPR knockout screens, creating a map of potential drug synergies with commonly used chemotherapeutics. To address the clear clinical need, we emphasized high-risk neuroblastoma cell lines, although several additional cancer and normal cell types were represented. We have made this dataset available to the research community as a resource to prioritize potential standard-of-care drug-drug synergies and to study drug mechanisms and resistance. We have used this resource to discover drug combinations with clinical potential, which we have demonstrated to be effective in vivo for doxorubicin and PRKDC inhibition using patient-derived xenograft models.

The observation that CRISPR knockout of a gene often mimics pharmacological targeting of the gene's protein product has now been widely exploited as a massively high-throughput proxy for single-agent drug screening (even when knockout does not mimic pharmacological targeting, it is often due to unconventional drug action, such as trapping an enzyme to DNA[74]). This has led to very large dependency maps[13], now profiling over 1000 cancer cell lines, which have been an immensely successful resource for drug repurposing[75] and drug discovery[76], especially in pediatric cancer[77]. This includes neuroblastoma, motivating for example the investigation of EZH2 inhibitors[78]. However, CRISPR knockout has not yet been extensively deployed to increase the throughput of drug combination discovery, although a few smaller-scale combinatorial knockout studies have been performed[13,32,79,80]. Some of the trends described in these previous studies were also evident in our data, for example, that cell line-specific synergies appear to be common, although we extend this idea to commonly used chemotherapeutics. Additionally, broad synergies, evident across a range of cell types are also common, which we were able to determine by screening an outgroup, a step typically overlooked in drug combination screening. However, even given these "too narrow" or "too broad" synergies, we still found many examples of synergies that appear to have context specificity in neuroblastoma, which was in part possible due to our statistical modeling. To promote the widespread use of these models, we have made these analytical tools available for future studies. Arguably, the major limitation of our study may be that, while it is on par with the largest drug combination screens ever performed, context specificity of synergies could likely be resolved in greater detail if screens were carried out using even larger numbers of cell lines in the future, or if data from many screens were aggregated. Interestingly, the proposed approach is highly scalable and effective and can be scaled up for many (if not most) cancers where the drug-drug combinatorial landscape remains almost entirely unexplored. Overall, CRISPR-drug combinatorial screens are effective for the discovery of potentially clinically relevant combinations with existing chemotherapeutics, which has the potential to impact patient care across a wide range of cancer types.

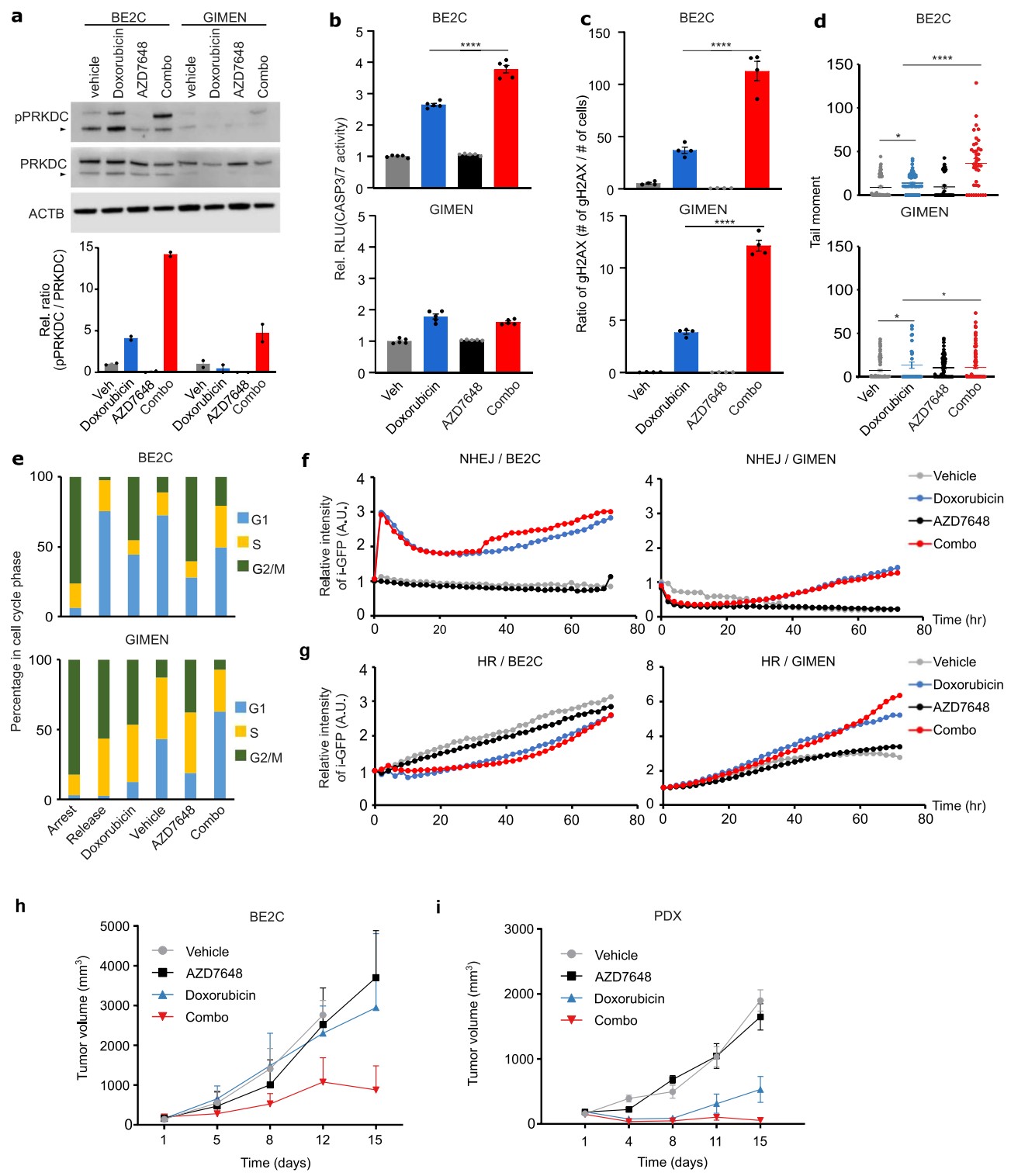

**Methods**

**Animals**

All murine experiments were done in accordance with a protocol (#615) approved by the Institutional Animal Care and Use Committee of St. Jude Children's Research Hospital. Around 5 weeks old female NSG mice (NOD.Cg-Prkdc scid Il2rg tm1Wjl /SzJ) were purchased from St Jude Children's Research Hospital Animal Research Resource and housed in pathogen-free conditions with food and water provided ad libitum. To establish SJNB14-PDX model, PDX tumor was finely minced with sterile scissors and blade in a sterile petri dish. ~50 µl of minced tumor tissue was subcutaneously engrafted on the right flank of NSG

mice. For generating BE2C xenograft, BE2C cells (5×10⁶/mouse) in 100 µl in Matrigel (Corning, 354230) were injected subcutaneously on the right flank of NSG mice.

**Cell culture and generation of Cas9 stably expressing cell lines**

18 cell lines and their associated Cas9-expressing cell lines (total 36) were cultured in the indicated culture condition (Supplementary Data Table 14, MHHNB11 (DSMZ, ACC157), BE2C (Easton@St.Jude, In-house), NGP (DSMZ, ACC676), KELLY(Sigma, 92110411), CHP212 (Shelat@St.Jude, In-house), GIMEN (DSMZ, ACC654), SKNAS (ATCC, CRL-2137), TGW (JCRB, JCRB0618), SKNFI (ATCC, CRL-2142), SKNS

**Fig. 6 | PRKDC inhibition represents a mechanistically plausible combination with doxorubicin in neuroblastoma, with evidence of synergistic activity in vivo. a** Western blot for phosphorylated PRKDC (active PRKDC) in BE2C and GIMEN cell lines. Results are quantified in the lower panel, where doxorubicin (0.4 $\mu$M in BE2C, 0.03 $\mu$M in GIMEN, same dose used in panels **a**–**g**) activated PRKDC compared to control (veh) whereas AZD7648 (3 $\mu$M, same dose used in panels **a**–**g**) inhibited PRKDC. The upper band represents the full-length protein and the N-terminus fragment of PRKDC is indicated by the triangle Data in lower panel presented as mean ± SEM, **$P = 0.00113$, unpaired $t$-test (two side), $n = 2$ per group, two independent. **b** Luminescent caspase 3/7 assay quantifying the level of apoptosis ($y$ axis) in vehicle, doxorubicin, AZD-7648, or combination ($x$ axis) treated cells. The results for BE2C are shown in the upper panel and GIMEN in the lower panel Data presented as mean ± SEM, ****$P < 0.0001$, unpaired $t$-test (two side), $n = 5$ per group, two independent. **c** Like (**b**) but for γH2AX foci from immunofluorescence assay. See Supplementary Data Table 12 for source data (Data presented as ratio of mean number of γH2AX over mean number of nuclei ± SEM, ****$P < 0.0001$, unpaired $t$-test (two side), $n = 4$ images per group, two independent). **d** Like (**b**) but for comet tail assay (Data presented as individual tail moments, *$P < 0.05$, ****$P < 0.0001$, unpaired $t$-test (two side), $n = 4$ images per group, two

independent). **e** Bar plot showing the percentage of cells ($y$ axis) in different phases of the cell cycle (colors), following either cell cycle arrest, 24 h following release, or treatment with vehicle, doxorubicin, AZD7648, or the combination for 72 h ($x$ axis). **f** Line plot showing the results of a cell-based assay for NHEJ using i-GFP quantifying the activity of NHEJ ($y$ axis) in BE2C and GIMEN cells over 72 h ($x$ axis), at 2 h intervals (Data presented as mean ± SEM, $n = 25$ of relative intensity of green per group (veh, doxorubicin, AZD7648, and combo), two independent). **g** Line plot showing the results of a cell-based assay for HR using i-GFP quantifying the activity of HR ($y$ axis) in BE2C (left panel) and GIMEN (right panel) cells over 72 h ($x$ axis) at 2 h intervals (Data presented as mean ± SEM, $n = 25$ of relative intensity of green per group (veh, doxorubicin, AZD7648, and combo), two independent). **h** Changes in tumor growth for BE2C xenografts (vehicle = 3, AZD7648 = 4, Doxorubicin = 5, Combo = 5). Data presented as mean ± SEM. Unpaired $t$-test (two-side) for comparison of doxorubicin with combination therapy: $P$-values: $3.5 \times 10^{-2}$ (day 8), $1.2 \times 10^{-2}$ (day 12), $1.6 \times 10^{-4}$ (day 15). **i** Changes in tumor growth for SJNB14 PDX xenografts (vehicle = 4, AZD7648 = 4, Doxorubicin = 5, Combo = 5). Data presented as mean ± SEM. Unpaired $t$-test (two-side) for comparison of doxorubicin with combination therapy: $P$-values: $5.4 \times 10^{-2}$ (day 11), $6 \times 10^{-5}$ (day 15). *Fig. 6a–i. Source data are provided as a Source Data file.

(Sigma, 86012802), SKES1 (Dyer@St.jude, In-house), SKMEL2 (Dyer@St.jude, In-house), RH30 (ATCC, CRL-2061), HCT116 (NCI), HEK293T (Chen@St.jude, In-house), AC16 (Millipore, SCC109), BJ-TERT (ATCC, CRL-4001), GM12878 (Easton@St.Jude, In-house)) and maintained in a mycoplasma-free condition. For CRISPR screens, Cas9 stably expressing cell lines were generated or obtained. Cas9 expressing SKNFI, TGW, SKNSH, SKES1, SKMEL2, RH30, HEK293T, BJ-TERT, AC16, and GM12878 were generated by transducing lentiviral Cas9-2A-Blast, followed by blasticidin selection. Cas9 expressing MHHNB11, BE2C, NGP, KELLY, CHP212, GIMEN, and SKNAS were provided by Dr. Adam Durbin. Cas9 expressing HCT116 was purchased from Horizon Discovery (Cat # Cas9-002). Cas9 activity in 18 Cas9-expressing cell lines were verified and it was over 85% on average using the Cas9 activity assay described at Method details.

### Generation of CRISPR KO lentiviral library
sgRNAs for the human CRISPR KO library were first designed using CRISPick, which designs and scores potential gRNAs based on several parameters from "Rule Set 3"[81,82]. The top 30 sgRNAs for each gene then underwent an additional round of filtering using in-house off-target analysis to identify highly unique sgRNAs. To balance library size with the statistical power of having multiple gRNAs, up to 6 sgRNAs per gene were selected for the library along with non-targeting controls making up ~10% of the final library. The sgRNA sequences are described in Supplementary Data Table 1. Library oligos were designed according to Sanson et al.[81]. The oligo pool was synthesized by TWIST Bioscience. Library amplification and Golden Gate cloning into the pLentiGuide-Puro backbone (Addgene #52963) were performed according to Sanson et al.[81]. The plasmid library was amplified and validated in the Center for Advanced Genome Engineering at St. Jude as described in the Broad GPP protocol. The only exception being the use of Endura DUOs electrocompetent cells. The St. Jude Hartwell Center Genome Sequencing Facility provided all NGS sequencing. Single end 100 cycle sequencing was performed on a NovaSeq 6000 (Illumina). Validation to check gRNA presence and representation was performed using calc_auc_v1.1.py (https://github.com/mhegde/) and count_spacers.py[83]. Viral particles were produced by the St. Jude Vector Development and Production laboratory. CRISPR KO screens were analyzed using Mageck-Vispr/0.5.7[84].

### Cas9 activity assay
Using a Cas9 activity assay kit (Cellecta, CRUTEST), Cas9 activity was measured by following the manufacture's protocol. Briefly,

Cas9-expressing cells were infected by CT-active [CT-A] or CT-background [CT-B] premade lentiviruses and maintained the infected cells for 10 days to avoid 100% confluency. After 10 days, the cells were analyzed by flow cytometry to measure the changes in GFP levels and Cas9 activity was determined (Supplementary Data Fig. 9 and Supplementary Data Table 15).

### Compounds and pharmacological profiling
Six standard drugs, cisplatin (CDDP, HY-17394), doxorubicin HCl(HY-15142), etoposide (HY-13629), topotecan HCl (HY-13768A), vincristine sulfate (VCR, HY-N0488), and all-trans retinoic acid (ATRA, HY-14649) were obtained from Medchemexpress (USA), and phosphoramide mustard (PM, D-18846) was obtained from Niomech IIT GmbH (Germany). JQAD1 was provided by Dr. Jun Qi (Dana-Farber Cancer Institute). As a broad-spectrum cell death compound (positive control for cell death), staurosporine (Medchemexpress, HY-15141) was used for evaluating cell viability. All compounds were reconstituted in DMSO, except CDDP. CDDP was reconstituted in normal saline or a mixture of DMSO and HCl (30 v:1 v). For pharmacological profiling, individual standard drugs were dispensed by Echo 650 (Labcyte) into white 384-well plates in a dose-dependent manner, followed by plating Cas9 expressing cells with desired numbers (500 or 1000 cells per well). After 3 days (CDDP, PM, doxorubicin, etoposide, topotecan, VCR) or 6 days (ATRA, JQAD1) incubation, CellTiter-Glo (Promega) assay was performed to determine viability and IC$_{20}$, IC$_{30}$, and IC$_{50}$ was calculated by fitting Hill Slope equation (Supplementary Data Fig. 10). An IC$_{20}$ to IC$_{30}$ was used for negative selection in our CRISPR screens. For the combinatorial drug screen, selected partner compounds, AZD7648 (PRKDC inhibitor, HY-111783), Panobinostat (HDAC2 inhibitor, HY-10224), dimethyl fumarate (KEAP1 inhibitor, HY-17363), and cabozantinib (MET inhibitor, HY-13016) were purchased from Medchemexpress.

### CRISPR screening
Our Cas9-expressing cell lines were infected with our SKOOL library at MOI (~0.3), followed by puromycin selection. In between days 10–12, once survived knockout cells reached at desired numbers to maintain representation, they were treated with individual standard drugs (IC$_{20-30}$) for 3 days (CDDP, PM, doxorubicin, etoposide, topotecan, VCR) or 6 days (ATRA and JQAD1). Genomic DNA was extracted using the PureLink genomic DNA kit (Invitrogen) and the sgRNA sequences were recovered by genomic PCR analysis, followed by deep sequencing using NovaSeq for paired-end minimum length 75 bp read (Illumina). Sequencing data were analyzed using MAGeCK-VISPR[84].

## Basic data analyses

UMAP plots were created using the M3C[85] package in R, which automatically selects the typically-user-defined UMAP plotting parameters using a Monte Carlo approach. Basic analysis of the CRISPR results was performed using MAGeCK-VISPR[84]. The visualization of cell line genomic features was created using ProteinPaint (https://proteinpaint.stjude.org/)[86]. All other basic analysis and statistical tests were performed using R version 4.0.2[87]. False discovery rates were estimated using the Benjamini and Hochberg method.

## Hierarchical Bayesian model to identify neuroblastoma selective hits and to borrow information across mechanistically related drugs

First, gRNAs acting as outliers were filtered using a Dixon outlier test. For each gene, this test was applied to log drug vs vehicle fold-changes of normalized read counts, and misbehaving gRNAs were removed at a nominal $P$-value threshold of 0.05. We filtered single gRNAs in the cases where the directionality of that single gRNA was different from the other 5 gRNAs targeting the same gene. We also filtered gRNAs with very low (<5) read counts in both the drug and treatment groups, as these produce very unstable fold change estimates. Drug vs vehicle fold changes were calculated for each remaining gRNA in each sample. These gRNA-level fold changes were then grouped by gene to estimate gene level fold changes and their associated standard error ($y_i$ and $SE^2_{y_i}$ respectively in Eq. 1 below).

The differential synergizing effect of a gene knockout and the potency of that effect in neuroblastoma was estimated using the following hierarchical Bayesian model:

$$y_i|d_i,x_i,\alpha,\beta,\sigma^2,\nu \sim t\left(\nu,\alpha_{d_i}+\beta_{d_i}x_i,\sqrt{SE^2_{y_i}+\sigma^2_{d_i}}\right) \quad (1)$$

Where $y_i$ is the normalized mean gRNA fold change calculated for each gene (typically the average of 6 independent gRNA fold change estimates). $d_i$ indicates the drug, $x_i$ indicates case/control status encoded as 0 or 1, $\alpha$ are the estimates of synergizing potency in the neuroblastoma group and $\beta$ are the estimates of the differential drug synergizing effect of a knockout between the neuroblastoma cell lines and the outgroup, $\nu$ and $\sigma^2$ represent the degrees-of-freedom and the variance of the t-distribution respectively. $SE^2_{y_i}$ is the standard error squared associated with each fold change estimate $y_i$, and the model formulation above allows these error estimates to be accounted for in the estimation of this model's parameters. Sharing of information across DNA-damaging agents was achieved using the following hierarchical structure, where the $\alpha$ and $\beta$ parameters for each drug are assumed to be drawn from shared parameterized Normal distributions:

$$\beta_{d_i}|\mu_\beta,\sigma^2_\beta \sim N\left(\mu_\beta,\sigma^2_\beta\right) \quad (2)$$

$$\alpha_{d_i}|\mu_\alpha,\sigma^2_\alpha \sim N\left(\mu_\alpha,\sigma^2_\alpha\right) \quad (3)$$

The $\mu_\alpha$ and $\mu_\beta$ parameters can then be interpreted as the shared sensitizing effect of a gene knockout on this entire group of drugs, described for example in Fig. 4i for DNA-damaging agents. Each of these hierarchical parameters are assigned a weakly informative prior:

$$\mu_\beta,\mu_\alpha,\sigma^2_\alpha,\alpha^2_\beta \sim N(0,5) \quad (4)$$

We used a gamma prior distribution for the degrees of freedom parameter of the t-distribution, with the following previously proposed shape and rate parameter values, which are suitable to

implement a model reasonably robust to outliers[88]:

$$\nu \sim Gamma\,(2,0.1) \quad (5)$$

For the joint model sharing information across the 6 DNA-damaging agents and $6 \times 18 = 108$ cell lines, the following indices were used:

$$d_i \in \{1,\dots,6\} \quad (6)$$

$$i = 1,\dots,108 \quad (7)$$

The description and equations above specifically describe the most complex model we used, jointly modeling the DNA damaging agents. In practice, we also employed several simpler iterations of this model. For example, independent models can be fit for each drug by dropping this model's hierarchical structure (i.e. Equations 2–4) and fitting the resulting model for each drug independently and the general drug sensitizing effect of a knockout, irrespective of neuroblastoma/outgroup status, can be estimated by dropping the model's $\beta$ parameter. The parameters were estimated using Hamiltonian Monte Carlo, implemented in the R package *rstan* (http://mc-stan.org/).

Finally, the gene set level analyses were performed by comparing the drug vs vehicle fold changes of genes in the relevant set (a pathway, a process, etc.) against a null distribution derived from a set of negative control genes (approach conceptually similar to Makrooni et al.)[89]. In this context, these negative control genes were created by randomly grouping the 400 non-targeting control gRNAs included in all screens into 66 negative control genes (6 gRNAs per gene). A non-parametric Wilcoxon rank sum test was then used to test the difference of the fold changes observed in each gene set against these negative control genes, which provides an estimate of the expectation under the null. Note that in this context the power of any gene set analysis approach to detect true positive associations will be restricted to groups of genes that are reasonably represented in the 655 gene library.

## Cisplatin-DNA adduct assay

Cells were treated with vehicle (normal saline or DMSO-HCl), cisplatin in normal saline, and cisplatin in DMSO-HCl for 24 h. Genomic DNA was extracted from treated cells and blotted in nitrocellulose membrane. The membrane was baked at 80 °C to immobilize blotted gDNA for 2 h. The baked membrane was blocked by TBST at room temperature (RT) for 30 min, followed by incubation of primary anti-cisplatin DNA adducts antibody (1:1000, Millipore, MABE416) at 4 °C overnight. The next day, the membrane was rinsed three times with TBST and incubated with secondary HRP-conjugated anti-rat at RT for 30 min. After rinsing three times, the membrane was imaged by Licor Odyssey XF to measure DNA adduct (Supplementary Data Fig. 11).

## Genetic validation of sensitized candidates

Three shRNAs per gene (PRKDC, HDAC2, KEAP1, and MET), were purchased (Sigma, Supplementary Data Table 14, PRKDC shRNA1 TRCN0000194985, shRNA2 TRCN0000195491, shRNA3 TRCN0000006258, HDAC2 shRNA1 TRCN0000004819, shRNA2 TRCN0000004823, shRNA3 TRCN0000195198, KEAP1 shRNA1 TRCN0000154656, shRNA2 TRCN0000154657, shRNA3 TRCN0000155340, MET shRNA1 TRCN0000040044, shRNA2 TRCN0000121087, shRNA3 TRCN0000121090, and non-targeting SMARTvector hEF1a-None (Horizon Discovery, VSC11723). Lentiviruses of each shRNA were produced at our institutional core facility, including non-targeting control. 10 or 11 cell lines were transduced by individual shRNA lentiviruses for 3 days and followed by puromycin selection to isolate the cells with the desired knockdown. Knockdown efficiency was verified by western blot analysis. We tested the

knockdown efficiency of individual shRNAs against *PRKDC, HDAC2, KEAP1*, and *MET* to select the most efficient one (TRCN0000194985 for *PRKDC*, TRCN0000004819 for *HDAC2*, TRCN0000154657 for *KEAP1*, and TRCN0000121087 for *MET*). Knockdown cells were treated with the corresponding drugs ($IC_{50}$ or 10 μM) for 3 days (doxorubicin/ PRKDC knockdown, JQAD1/HDAC2 knockdown, topotecan/KEAP1 knockdown, CDDP/MET knockdown) to test whether the potency of four drugs was increased.

## Dense drug-drug synergy screening assays using high throughput robotic handling

Doxorubicin, AZD7648, cisplatin, cabozantinib, topotecan, dimethyl fumarate, JQAD1, and panobinostat were added into 384-well Perkin Elmer Culture plates (Perkin Elmer #6007688) using Labcyte Echo 555 and 655 T acoustic liquid dispensers moving a total volume of 80 nL into each well to generate desired combinations with an equivalent final DMSO concentration of 0.2%. Each assay plate contained ten-point dose-response curve with 1:3 dilution intervals for each compound, three replicates of the pairwise drug combination matrices, three replicates of each compound alone, and twelve replicates of each control (DMSO and 20 μM staurosporine to represent 0% and 100% cell death, respectively). Seeding densities were determined a priori by growth assays for each of the 18 cell lines. Cell lines were plated into assay plates in volumes of 40 μL using a Multidrop Combi reagent dispenser to reach desired the final desired concentrations, and then settled for 20 s at $200 \times g$ in a Sorvall Legend XTR centrifuge. Plates were then incubated at 37 °C in 5% $CO_2$ for 72 h in a High Resolution Biosolutions Steristore incubator. After incubation the plates were moved to a Liconic STX220 incubator, at 37 °C in 5% $CO_2$, integrated onta an Agilent BioCell, to determine viability. The plates were each placed at room temperature for 20 min before viability assessment. 25 μL of CellTiter-Glo reagent (Promega #G9241) was added to each well using a Multidrop Combi to measure viability, and the plates were incubated for an additional 20 min at room temperature. Luminescence was then measured with an EnVision 2102 Multilabel Plate Reader.

## Analysis of dense drug-drug synergy screening data

The raw luminescent data was imported into R. Background-subtracted values in RLU were assigned to the appropriate drugs and concentrations. All replicates were normalized to the mean of their respective inter-plate controls (vehicle for 0% cell death and staurosporine for 100% cell death). Normalized drug-only data were fit with log-logistic regression to produce dose-response curves using the DR4PL package[90,91]. Matrices of the percent cell death values were constructed using means of normalized data from each of the replicates per treatment combination as input. From these normalized values, synergy scores were calculated for all tested concentration combinations, using the Zero Interaction Potency (ZIP) model implemented using the *SynergyFinder* package in R. The resulting synergy matrices were used to extract the highest- and lowest-scoring concentration pairs to represent the most significant synergy and antagonism.

## AZD7648 (PRKDCi) and Doxorubicin treatment in SJNB14-PDX and BE2C Xenograft models

Doxorubicin was purchased from Selleckchem (Selleckchem, S1208). AZD7648 were purchased from Chemietek (Chemietek, CT-A7648) and formulated in 0.5% hydroxypropyl methylcellulose (Sigma, H3785-100)/0.1% Tween80 (Sigma, P4780–500 mL) (HPMC/T) for oral gavage. Tumor size was measured with electronic calipers. The tumor volume was calculated using the formula $\pi/6 \times d^3$, where d is the mean of two diameters taken at right angles. The tumor volume and mice weight were measured twice a week. When tumor size reached up to ~100–200 mm³, the animals were randomized into four groups (*n* = 4–5

mice per group). Mice were treated with vehicle (HPMC/T), doxorubicin (0.75 mg/kg, intraperitoneal, twice weekly), and AZD7648 (50 mg/kg, twice/day, oral gavage every day and the time between the morning and evening doses was 8 h) and combination of doxorubicin and AZD7648 for two weeks. On the day of doxorubicin treatment, doxorubicin was dosed 1 h after the morning dosing of AZD7648. The humane endpoints were monitored and decided by ARC (Animal Resources Center) staff of St. Jude and informed to euthanize the mice. All measured tumor volume was reported at Source Data file provided in this study. The maximal tumor burden permitted was 20% of mouse body weight or 4000 mm³, and in our experiments, the maximal tumor burden was not exceeded. For therapy studies in subcutaneous xenograft mouse models, the mice were euthanized through $CO_2$ inhalation with 3 Liters/min in the mouse cage and followed by cervical dislocation when the tumor volume reached or exceeded 4000 mm³ (~20% body weight which was in a range 20–25 g) or mice became moribund.

## Western blot analysis

Cells were treated with vehicle (DMSO) or drug for 72 hrs. Total proteins were extracted by using a modified RIPA buffer (HEPES, NaCl, EDTA, PI cocktail tablet, PPi cocktail tablet, PMSF, DTT). Total 30 μg of proteins were resolved at gradient gels (Biorad). Resolved proteins were transferred to nitrocellulose membrane by using iBlot (Invitrogen). The membrane was blocked by TBST at room temperature (RT) for 30 min, followed by primary antibodies (1:1000, anti-Cas9 (Cell signal technology, 14697 S), anti-Actin (Sigma, A2228), anti-PRKDC (Cell signal technology, 38168 S), anti-phospho PRKDC (S2056) (Cell signal technology, 68716 S), anti-HDAC2 (Santacruz Biotech, sc-9959), anti-MET (Cell signal technology, 8198 S), anti-KEAP1 (Cell signal technology, 8047 S)) at 4 °C for overnight. The next day, the membrane was rinsed three times with TBST and incubated with secondary antibodies at RT for 1 h. After rinsing three times, the membrane was imaged by Licor Odyssey XF to measure the level of target protein levels.

## Immunofluorescent staining

72 h after treatment with vehicle (DMSO) or drug, the cells were fixed with 4% paraformaldehyde at RT for 10 min and rinsed with 1× PBS three times. The rinsed cells were permeabilized with 0.1% Triton-X 100 in 1× PBS at RT for 10 min, blocked with 0.5% fetal bovine serum, 0.01% Triton-X 100 in 1× PBS at RT for 30 min. After blocking, the cells were incubated with mouse anti-γH2AX (1:500, Cell signal technology, 80312 S) or rabbit anti-phospho PRKDC (S2056) (1:200, Cell signal technology, 68716 S) at 4 °C overnight. The next day, the cells were rinsed with 1× PBS three times and incubated with anti-mouse Alexa488 or anti-rabbit Alexa594 at RT for 60 min. Fluorescent images were taken by Nikon Ti Eclipse and analyzed by CellProfiler (https:// cellprofiler.org).

## Caspase 3/7 assay

Apoptosis was measured by luminescent caspase 3/7 assay (Promega). Briefly, 72 hrs after treatment with vehicle (DMSO) or drug in BE2C and GIMEN in 96 well plates, 100 μL of caspase 3/7 reagent was directly added to the cell plates. The cells were incubated in the dark at room temperature for 60 min, followed by reading luminescence using a CLARIOstar plate reader (BMG Labtech).

## Cell cycle arrest and drug treatment

BE2C and GIMEN were incubated with serum free medium for 48 h to arrest G0/G1. 48 h after starvation, the cells were released by adding fresh complete growth medium. BE2C and GIMEN were treated with 100 ng/mL of nocodazole (NOC, Medchemexpress, R17934) for 18 h to arrest G2/M phase. 18 h after treatment, the cells were released by washing out NOC three times and replaced with fresh complete growth medium. 24 h after release of each synchronization, the cells were

treated with vehicle (DMSO) or drugs for 72 h. The treated cells were harvested at 1000 RPM for 10 min, fixed with 70% ethanol at −20 C for 2 h, then washed with 1x PBS, and staining with propidium iodide (50 ug/mL, BD PI/RNase staining buffer, BD Biosciences, 550825) for FACS.

## NHEJ, HR,and PRKDC-related assays

Non-homologous end joining (NHEJ) and homologous recombination (HR) in BE2C and GIMEN neuroblastoma cell lines was visualized by using a custom cell-based kit for NHEJ (Topogen, Cat # DR5000A) and HR (Topogen, Cat # DR3000A). Following the manufacturer's guide, the cells were transfected with a modified GFP reporter, along with a plasmid encoding I-*Sce*I, or empty vector by using TransIT (Mirus, Cat # MIR5400) in 6-well plates. 24 h after incubation, the cells were trypsinized, and replated into 12-well plates for single treatment of doxorubicin, AZD7648, vehicle (DMSO), and combination treatment of doxorubicin and AZD7648. The changes in GFP of cells in 12-well plates were monitored by using IncuCyte SX5 (Sartorius) for 72 h.

## Comet assay

Comet assay was performed according to the manufacturer's protocol (Abcam, ab238544). Vehicle or drug treated BE2C and GIMEN cells were harvested and resuspended in cold 1x PBS at 100 cells/uL. The cells were mixed with comet agarose (1:10 volume ratio) and transferred to slides. The slides were lysed with alkaline buffer (0.3 M NaOH, 1 mM EDTA) at 4 °C for 60 min. Slides were then subjected to electrophoresis at 35 V for 30 min in alkaline buffer, then fixed with 70% cold ethanol for 5 min. After fixation, the slides were stained with Vista DNA dye (Abcam, ab238544) to visualize DNA/nuclei. The images of the slides were taken by Nikon Ti Eclipse and analyzed by OpenComet tool (https://cometbio.org/).

## Statistics and reproducibility

Sample size for each experiment is indicated in the legend although no statistical method was used to predetermine sample size. All experiments were conducted on at least 2–3 independent biological replicates. Measurements were biological replicate samples. For in vitro experiments, cell cultures are randomly assigned to each experimental group. For tumor-growth measurement, mice were randomly assigned to each experimental group with various treatments. Data are presented as mean ± SEM from at least three biological replicates unless otherwise stated. For comparisons of two experimental groups, unpaired t-test (two-side), or non-parametric Wilcoxon rank sum test was used. Statistical significance is represented by asterisks corresponding to $*p < 0.05$, $**p < 0.01$, and $****p < 0.0001$. GraphPad Prism software (version 9.0) or R package (4.2) was used to generate graphs and perform statistical analyses.

## Reporting summary

Further information on research design is available in the Nature Portfolio Reporting Summary linked to this article.

# Data availability

The raw sequencing data was generated for this study and have been deposited in GEO (GSE223991). Summarized gRNA and gene level data are included as Supplementary Data Tables. Detailed information on materials is in the Supplementary Data Table 14. All remaining data can be found in the Article, Supplementary, and Source Data files. Source data are provided with this paper (Source data file.xlsx). Source data are provided with this paper.

# Code availability

The code to reproduce the analyses have been deposited on Open Science Framework (https://osf.io/d9xgn/) and https://stjude.shinyapps.io/CASAVA/.

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

## Acknowledgements

P.G. is supported by an NIGMS R35 award [R35GM138293] an R01 grant from NCI [R01CA260060]; K99/R00 [R00HG009679] from NHGRI; P.G, A.D.D., and J.Y. also receive support from ALSAC. Funding for open access charge: NIH. A.D.D. and the St. Jude Center for Advanced Genome Engineering is funded by the NCI P30 CA021765. J.Y. was partly supported by American Cancer Society-Research Scholar (130421-RSG-17-071-01-TBG, J.Y.) and National Cancer Institute (1R01CA229739-01, 1R01CA266600-01A1). A.D.D. is supported by funding from the NCI (K08CA245251-01A1), the Curesearch For Children's Cancer Foundation, the Alex's Lemonade Stand Foundation, Hyundai Hope on Wheels Foundation, V Foundation for Cancer Research and the Rally Foundation for Childhood Cancer Research. The authors thank Sarah K. August for assisting with preparing the scientific illustrations. The content is solely the responsibility of the authors and does not necessarily represent the official views of the National Institutes of Health. The authors have declared that no conflict of interest exists.

## Author contributions

P.G. conceived and directed the study. H.M.L. performed the screens and the mechanistic work. W.C.W. and P.G. analyzed the high throughput data. P.G., H.M.L. and W.C.W. drafted the paper. H.M.L., W.C.W., S.M., A.D and P.G. designed the study. J.F., S.S. and J.Y. performed the in vivo work. W.C.W., M.P., J.L., D.C. and T.C. performed the combination screening. M.P., J.F., S.S., S.N., I.D., Y.K., S.M., J.Q., J.E. and A.D. performed additional experimental work. R.H.C., Y.Z., X.L. and J.A.S. performed additional analytical work.

## Competing interests

The authors declare no competing interests.

## Additional information

[1]Department of Computational Biology, St. Jude Children's Research Hospital, Memphis, TN 38105, USA. [2]Department of Chemical Biology, St. Jude Children's Research Hospital, Memphis, TN 38105, USA. [3]Department of Surgery, St. Jude Children's Research Hospital, Memphis, TN 38105, USA. [4]Division of Molecular Oncology, Department of Oncology, St. Jude Children's Research Hospital, Memphis, TN 38105, USA. [5]Center for Advanced Genome Engineering, St. Jude Children's Research Hospital, Memphis, TN 38105, USA. [6]Department of Cell and Molecular Biology, St. Jude Children's Research Hospital, Memphis, TN 38105, USA. [7]Department of Cancer Biology, Dana-Farber Cancer Institute, Boston, MA, USA. [8]Department of Medicine, Harvard Medical School, Boston, MA, USA. [9]Department of Pathology and Laboratory Medicine, College of Medicine, The University of Tennessee Health Science Center, Memphis, TN 38163, USA. [10]These authors contributed equally: Hyeong-Min Lee, William C. Wright. ✉e-mail: Jun.Yang2@stjude.org; Adam.Durbin@stjude.org; Paul.Geeleher@stjude.org

