## [Peer Review File · Nature Communications]

Reviewers' Comments:

Reviewer #2:

Remarks to the Author:

This is a revised version of a manuscript previously submitted to another journal.

The authors report on a large-scale targeted CRISP knock-out screen, analyzing the effect of knock-out of targetable genes on chemotherapeutic effects in 10 neuroblastoma cell lines as well as control cell lines. The results concern the discovery of some drugs which sensitize neuroblastoma cell lines to standard of care chemotherapy, suggesting potential new combinations for high-risk neuroblastoma. An important aspect of the work is the data scope with its availability via a graphical, web-based interface.

The new version of the manuscript provides additional information of the choice and genetic background of cell lines, and on the selected drugs. Importantly methodological aspects of the CRIPSR knock-out screen are included in more detail, addressing aspects of quality control (such as information on the non-targeting, core-fitness and pan-essential genes). Figures and supplemental material reflect the previously suggested changes.

Altogether this revised version of the manuscript addresses all previously raised issues.

I think this manuscript includes new findings of interest, descriptions of a methodology which is certainly of interest to a wide readership, and a robust and sustainable data resource .

Reviewer #3:

Remarks to the Author:

The authors have invested significant effort in addressing the points I raised in the previous round of review, especially those related to the lack of robust quality control and reproducibility in their screening process. Consequently, their manuscript has become considerably more robust, thereby supporting their final claims with greater rigor. This study is poised to make an excellent contribution to the field and is bound to pique the interest of Nature Communications' readers.

Reviewer #4:

Remarks to the Author:

Lee et al have conducted a CRISPR screen to 655 known druggable genes against 18 different cell lines and 8 different chemotherapeutic drugs (doxorubicin, cisplatin, phosphoramidate, etoposide, topotecan, vincristine, all trans retinoic acid and JQAD1 in over 94,000 unique combinations. 10 of the cell lines are neuroblastoma, a devastating childhood cancer, 4 lines from other cancers and 4 cell lines from normal cells. For validation they include knock out of genes for which the specific drug target is known, for example CX5461 and Top2B. They generate a wealth of data that they analyse in the manuscript using various approaches. Some of the key findings are that

1. The experiments clustered, in large part, by cell line rather than drug type.
2. Knockout of MCL1 sensitized SKMEL2 melanoma cells to 8 different drugs
3. Several of the findings had clear biologic rationale for example PARP1 was the top hit for sensitizing to the Top1 inhibitor topotecan.
4. They also find that PRKDC was the most potent sensitizer to doxorubicin, and BCL2L1 to cisplatin and MCL1 was highly ranked for a range of DNA damaging drugs. These observations have clear relevance to drug development.
5. PRKDC knock out also preferentially sensitized adrenergic neuroblastoma cell lines.
6. Top candidates were confirmed using shRNA and where possible pharmacological inhibition.
7. Focusing on PRKDC, the assay for protein activation (autophosphorylation at S2056) following doxorubicin treatment in 2 neuroblastoma cell lines, PRKDC knock out strongly synergised with doxorubicin in BE2C cells but not GIMEN cells.
8. They go on to test this combination in a mouse model of neuroblastoma and a PDX model. In

both cases the combination of DNA-PK inhibitor and doxorubicin had a pronounced effect on tumour volume.

Although other large combinational drug screens have been published (references 73 and 33). The difference is that this study uses more cancer and non-cancer cell lines, is the first to focus on pediatric tumours and the authors claim is less labour intensive than previous methods.

The authors have made their results available to the research community via a database (line 404)

Overall, I think this is a strong study that will be very useful to the research community.

I have a few comments:

Point 1: In lines 514-520, the authors note that 2056 phosphorylation increased with doxorubicin and the DNA-PK inhibitor AZD7648. The authors may wish to reconsider their conclusions in this section to acknowledge that DNA-PKcs S2056 phosphorylation is a documented effect of apoptosis, which fits with their findings. The enhanced H2AX phosphorylation in these cells is also likely due to apoptosis.

Please see:

Bipasha Mukherjee 1, Chase Kessinger, Junya Kobayashi, Benjamin P C Chen, David J Chen, Alope Chatterjee, Sandeep Burma: DNA-PK phosphorylates histone H2AX during apoptotic DNA fragmentation in mammalian cells. *DNA Repair (Amst)*. 2006 May 10;5(5):575-90.
doi: 10.1016/j.dnarep.2006.01.011.Epub 2006 Mar 29. PMID: 16567133.

Point 2: Do the authors see evidence of DNA fragmentation in their treated cells? Did they see a subG1 peak in the flow cytometry assays or increased TUNEL assay staining?

Point 3: Also, DNA-PKcs is cleaved by Caspase 3 during apoptosis to generate bands an N-terminal fragment of approximately 250 kDa and 150/120 kDa C terminal fragments. This is likely the smaller, S2056 cross reacting band seen in Figure 6a.

Please see: DNA-dependent protein kinase catalytic subunit: a target for an ICE-like protease in apoptosis. Song Q, Lees-Miller SP, Kumar S, Zhang Z, Chan DW, Smith GC, Jackson SP, Alnemri ES, Litwack G, Khanna KK, Lavin MF.*EMBO J*. 1996 Jul 1;15(13):3238-46.PMID: 8670824

Point 4: It seems odd that the GIMEN cells are not showing DNA-PKcs 2056 phosphorylation in response to doxorubicin. Do the authors know that the drug is getting into the cells? For example do they have drug resistance? Does IR or other small molecule DNA damaging agents induce 2056 phosphorylation?

Point 5: Figure 6 is very small. It would be helpful to the reader if it could be made larger so it is easier to read.

Point 6: The authors state (line 614) that gene knock out often mimics pharmacological inhibition. This may be true in general but there are several important exceptions. For example some PARP inhibitors are thought to act by trapping the inhibited enzyme on the DNA, thus preventing other repair pathways from proceeding

see Murai et al,

Cancer Res. 2012 Nov 1; 72(21): 5588–5599.

doi: 10.1158/0008-5472.CAN-12-2753

This should be mentioned as a caveat in the discussion.

Dear Dr. Clancy and Reviewers,

We are glad to see that most of the original Reviewer comments have now been addressed. We have included a point-by-point response to the final Reviewer comments below.

(Line numbers quoted below refer to the tracked-revised manuscript unless otherwise stated.)

REVIEWER #2

This is a revised version of a manuscript previously submitted to another journal. The authors report on a large-scale targeted CRISP knock-out screen, analyzing the effect of knock-out of targetable genes on chemotherapeutic effects in 10 neuroblastoma cell lines as well as control cell lines. The results concern the discovery of some drugs which sensitize neuroblastoma cell lines to standard of care chemotherapy, suggesting potential new combinations for high-risk neuroblastoma. An important aspect of the work is the data scope with its availability via a graphical, web-based interface. The new version of the manuscript provides additional information of the choice and genetic background of cell lines, and on the selected drugs. Importantly methodological aspects of the CRIPSR knock-out screen are included in more detail, addressing aspects of quality control (such as information on the non-targeting, core-fitness and pan-essential genes). Figures and supplemental material reflect the previously suggested changes. Altogether this revised version of the manuscript addresses all previously raised issues. I think this manuscript includes new findings of interest, descriptions of a methodology which is certainly of interest to a wide readership, and a robust and sustainable data resource.

Response to Reviewer #2 comments

We thank the reviewer for their additional comments and positive assessment of the paper.

REVIEWER #3 (original Ref)

Reviewer #3 comments

The authors have invested significant effort in addressing the points I raised in the previous round of review, especially those related to the lack of robust quality control and reproducibility in their screening process. Consequently, their manuscript has become considerably more robust, thereby supporting their final claims with greater rigor. This study is poised to make an excellent contribution to the field and is bound to pique the interest of Nature Communications' readers.

Response to Reviewer #3 comments

We thank the reviewer for their extremely helpful comments, which have substantially improved the paper.

REVIEWER #4 (New ref) – DNA-PKi

Reviewer #4 General comments

Lee et al have conducted a CRISPR screen to 655 known druggable genes against 18 different cell lines and 8 different chemotherapeutic drugs (doxorubicin, cisplatin, phosphoramidate, etoposide, topotecan, vincristine, all trans retinoic acid and JQAD1 in over 94,000 unique combinations. 10 of the cell lines are neuroblastoma, a devastating childhood cancer, 4 lines form other cancers and 4 cell lines from normal cells. For validation they include knock out of genes for which the specific drug target is known, for example CX5461 and Top2B. They generate a wealth of data that they analyse in the manuscript using various approaches. Some of the key findings are that

- 1. The experiments clustered, in large part, by cell line rather than drug type.*
- 2. Knockout of MCL1 sensitized SKMEL2 melanoma cells to 8 different drugs*
- 3. Several of the findings had clear biologic rationale for example PARP1 was the top hit for sensitizing to the Top1 inhibitor topotecan.*
- 4. They also find that PRKDC was the most potent sensitizer to doxorubicin, and BCL2L1 to cisplatin and MCL1 was highly ranked for a range of DNA damaging drugs. These observations have clear relevance to drug development.*
- 5. PRKDC knock out also preferentially sensitized adrenergic neuroblastoma cell lines.*
- 6. Top candidates were confirmed using shRNA and where possible pharmacological inhibition.*
- 7. Focusing on PRKDC, the assay for protein activation (autophosphorylation at S2056) following doxorubicin treatment in 2 neuroblastoma cell lines, PRKDC knock out strongly synergised with doxorubicin in BE2C cells but not GIMEN cells.*
- 8. They go on to test this combination in a mouse model of neuroblastoma and a PDX model. In both cases the combination of DNA-PK inhibitor and doxorubicin had a pronounced effect on tumour volume.*

Although other large combinational drug screens have been published (references 73 and 33). The difference is that this study uses more cancer and non-cancer cell lines, is the first to focus on pediatric tumours and the authors claim is less labour intensive than previous methods. The authors have made their results available to the research community via a database (line 404)

Overall, I think this is a strong study that will be very useful to the research community.

Response to Reviewer #4 General comments

We thank the reviewer for their kind words about our study overall. We have responded point-by-point below to their specific comments.

Reviewer #4 comment #1

Point 1: In lines 514-520, the authors note that 2056 phosphorylation increased with doxorubicin and the DNA-PK inhibitor AZD7648. The authors may wish to reconsider their conclusions in this section to acknowledge that DNA-PKcs S2056 phosphorylation is a documented effect of apoptosis, which fits with their findings. The enhanced H2AX phosphorylation in these cells is also likely due to apoptosis.

Please see:

Bipasha Mukherjee 1, Chase Kessinger, Junya Kobayashi, Benjamin P C Chen, David J Chen,

Aloke Chatterjee, Sandeep Burma: DNA-PK phosphorylates histone H2AX during apoptotic DNA fragmentation in mammalian cells. DNA Repair (Amst). 2006 May 10;5(5):575-90. doi: 10.1016/j.dnarep.2006.01.011.Epub 2006 Mar 29. PMID: 16567133.

Response to Reviewer #4 comment #1

We thank the reviewer for this excellent point, which does indeed provide further context for some of the trends we see when PRKDC inhibitors are combined with doxorubicin in our neuroblastoma cells. As the reviewer has suggested, we have now included this reference and commented on this important detail that phospho-DNA-PKcs has also been documented to emerge as a consequence of apoptosis (lines 545-547). This adds an important additional layer to the interpretation of these data, and we thank the reviewer for graciously highlighting this important idea.

Reviewer #4 comment #2

Point 2: Do the authors see evidence of DNA fragmentation in their treated cells? Did they see a subG1 peak in the flow cytometry assays or increased TUNEL assay staining?

Response to Reviewer #4 comment #2

We agree with the reviewer that DNA fragmentation is an important consideration that would support our conclusions. Indeed, we see evidence of DNA fragmentation in our treated cells, which we determined by a comet tail assay. This is shown in Fig. 6d for the combination-sensitive BE2C cell line and the comparatively resistant GIMEN cell line. DNA-fragmentation is much higher in BE2C doxorubicin-treated cells and is heavily potentiated by the addition of AZD7648, which was not the case for GIMEN.

Reviewer #4 comment #3

Point 3: Also, DNA-PKcs is cleaved by Caspase 3 during apoptosis to generate bands an N-terminal fragment of approximately 250 kDa and 150/120 kDa C terminal fragments. This is likely the smaller, S2056 cross reacting band seen in Figure 6a.

Please see: DNA-dependent protein kinase catalytic subunit: a target for an ICE-like protease in apoptosis. Song Q, Lees-Miller SP, Kumar S, Zhang Z, Chan DW, Smith GC, Jackson SP, Alnemri ES, Litwack G, Khanna KK, Lavin MF.EMBO J. 1996 Jul 1;15(13):3238-46.PMID:8670824

Response to Reviewer #4 comment #3

We agree with the reviewer that we should have provided more detail on these different bands appearing on Fig. 6A. The upper band represents the full length PRKDC protein, and the lower band represents the 250kDa N terminal fragment, which we have now clearly stated in the figure legend (lines 577-582). We thank the reviewer for helping us provide more clarity on this figure.

Reviewer #4 comment #4

Point 4: It seems odd that the GIMEN cells are not showing DNA-PKcs 2056 phosphorylation in

response to doxorubicin. Do the authors know that the drug is getting into the cells? For example do they have drug resistance? Does IR or other small molecule DNA damaging agents induce 2056 phosphorylation?

Response to Reviewer #4 comment #4

We agree with the reviewer that this is worthy of further exploration. Firstly, both doxorubicin and AZD-7648 are certainly getting to the cells – for all experiments relevant to Fig. 6, doxorubicin was treated at an approximate IC₅₀ in both cell lines (i.e., we are using the drug at a concentration where there is a strong effect on cell viability). In the previous submission, we speculated these differences in e.g., ser-2056 phosphorylation between GIMEN and BE2C were related to differences in the relative usage of the different DNA-repair pathways. We have now added new data, which are also consistent with this argument. Specifically, we have repeated the NHEJ assays (previously Fig. 6f; now moved to Supplementary Figure 8), collecting the data across a denser set of time points, which is again consistent with a greater induction of NHEJ in BE2C, compared to GIMEN. In addition to this, we have assayed the relative usage of the homologous recombination (HR) pathway in these two different cell lines (see new revised Fig. 6f, g). Indeed, the induction of HR is much more pronounced in GIMEN, suggesting that GIMEN relies on HR, but BE2C much more heavily relies on NHEJ, which provides a plausible explanation for the greater synergy observed in BE2C when we target NHEJ using PRKDC inhibition.

Reviewer #4 comment #5

Point 5: Figure 6 is very small. It would be helpful to the reader if it could be made larger so it is easier to read.

Response to Reviewer #4 comment #5

We thank the reviewer for highlighting this shortcoming. We have now enlarged this figure and improved the sizing of Figure 6 in general.

Reviewer #4 comment #6

*Point 6: The authors state (line 614) that gene knock out often mimics pharmacological inhibition. This may be true in general but there are several important exceptions. For example some PARP inhibitors are thought to act by trapping the inhibited enzyme on the DNA, thus preventing other repair pathways from proceeding
see Murai et al,
Cancer Res. 2012 Nov 1; 72(21): 5588–5599.
doi: 10.1158/0008-5472.CAN-12-2753
This should be mentioned as a caveat in the discussion.*

Response to Reviewer #4 comment #6

We agree with the reviewer that these additional details of the circumstances when pharmacological targeting can phenocopy CRISPR knockout are important. We have now expanded on this important caveat in the Discussion (lines 635-637).

Overall, we thank the Reviewers for their detailed and extremely helpful comments. We are deeply grateful for the efforts of the Reviewers and Editor in working with us to sharpen our message and better present our findings.

Sincerely,

Paul Geeleher, PhD
Dept. Computational Biology
St Jude Children's Research Hospital,
Memphis TN,
USA.
paul.geeleher@stjude.org

Adam Durbin, MD, PhD
Dept. Oncology,
St. Jude Children's Research Hospital,
Memphis TN,
USA
adam.durbin@stjude.org

Jun Yang, PhD
Dept. Surgery
St Jude Children's
Research Hospital,
Memphis TN,
USA
jun.yang2@stjude.org